# From Forecasting to Planning: Policy World Model for Collaborative State-Action Prediction

**Zhida Zhao** [*]   **Talas Fu** [*]   **Yifan Wang**   **Lijun Wang** [†]   **Huchuan Lu**
Dalian University of Technology
{770153907, oyontalas}@mail.dlut.edu.cn
{wyfan, ljwang, lhchuan}@dlut.edu.cn

## Abstract

Despite remarkable progress in driving world models, their potential for autonomous systems remains largely untapped: the world models are mostly learned for world simulation and decoupled from trajectory planning. While recent efforts aim to unify world modeling and planning in a single framework, the synergistic facilitation mechanism of world modeling for planning still requires further exploration. In this work, we introduce a new driving paradigm named Policy World Model (PWM), which not only integrates world modeling and trajectory planning within a unified architecture, but is also able to benefit planning using the learned world knowledge through the proposed action-free future state forecasting scheme. Through collaborative state-action prediction, PWM can mimic the human-like anticipatory perception, yielding more reliable planning performance. To facilitate the efficiency of video forecasting, we further introduce a parallel token generation mechanism, equipped with a context-guided tokenizer and an adaptive dynamic focal loss. Despite utilizing only front camera input, our method matches or exceeds state-of-the-art approaches that rely on multi-view and multi-modal inputs. Code will be released at https://github.com/6550Zhao/Policy-World-Model.

## 1   Introduction

Driving world models have recently garnered growing research interest, due to their capacity to simulate future environmental states, enabling autonomous systems to anticipate complex traffic dynamics and enhance decision-making safety [1, 2]. This trend is particularly evident in video-based paradigms, driven by the accessibility of large-scale video datasets and the breakthroughs of the video generation technique [3–5]. Despite remarkable progress having been achieved, existing world model based autonomous driving methods  [6–10] mostly operate in a decoupled manner (Figure 1 (a)), *i.e.*, the world models aim to predict the next state and the associated reward but cannot directly perform trajectory planning. A separate policy model is still required for perceiving and planning. As a result, the full potential of the world model in autonomous driving is largely restricted [11].

Several concurrent works have made the initial attempt towards integrating both world modeling and planning in a unified autoregressive model [12, 11], where interleaved image and action token sequences are generated through the next-token prediction manner. Although the two tasks are jointly learned in these works, they are still independently conducted (Figure 1 (b)), where world modeling prioritizes high-fidelity next frame prediction with photo-realistic details conditioned on input actions, and trajectory planning is performed through an end-to-end mapping from the visual observation to output actions without explicitly leveraging the learned world model. As such, the world modeling and trajectory planning are only unified in terms of model architecture and prediction

---

[*]Equal contribution
[†]Corresponding author

39th Conference on Neural Information Processing Systems (NeurIPS 2025).

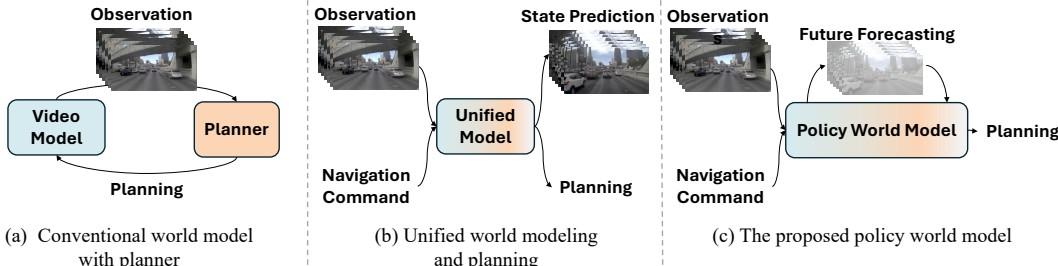

(a) Conventional world model
with planner     (b) Unified world modeling
and planning     (c) The proposed policy world model

Figure 1: Comparison of video world models for autonomous driving (a) Conventional video world models [6–10] typically serve as data engines for stimulation in pixel space. (b) Unified world models [11–13] perform video generation and planning as separate tasks. (c) Our proposed policy world model performs planning based on the learned world knowledge.

manner, while their synergistic mechanism remains under-explored. It is still unknown whether and how this unification can further benefit autonomous driving.

In light of the above observations, we propose a new driving world model which not only unifies world modeling and trajectory planning in a cohesive architecture, but is also able to leverage the learned world knowledge to enhance planning efficacy (Figure 1 (c)). Therefore, we name our model the **Policy World Model (PWM)**. When human drivers plan, they frequently imagine possible future environment states to anticipate potential hazards. To mimic this "Anticipatory Perception" maneuver, PWM performs trajectory planning with an action-free future forecasting framework. Specifically, PWM is pre-trained to acquire world modeling ability through auto-regressive video generation on unlabeled video sequences. During fine-tuning and inference, given current and historical video frames, it first generates textual description to understand the current environment and then rolls out plausible future states via video generation based on the learned world knowledge. The current action is finally predicted by considering the generated description and forecasted future states as multi-modal rationales. The above procedure is implemented by an end-to-end auto-regressive Transformer, ensuring seamless collaboration between perception, prediction, and planning. Compared to conventional world models, PWM acting as a driving policy can fully unleash its world knowledge to directly perform decision-making, rather than relying on model-based policy searching. In addition, the action-free video forecasting framework eliminates dependency on action-labeled data, not only ensuring training scalability but also allowing more flexible future state rolling out.

To improve the efficiency of video forecasting, we enable parallel generation of all tokens within a single frame, which allows video synthesis through next-frame prediction and substantially accelerates the forecasting process. For this purpose, we employ a compact image token representation (28 tokens per image via context-guided compression and decoding), ensuring both efficiency and coherent visual generation. To ensure video generation quality, we design a novel dynamic focal loss for training PMW to focus on temporally varying image regions. The combination of the above innovations allows PWM to generate high-quality future video frames at a reasonable computational overhead. We evaluate PWM on the popular benchmarks including nuScenes [14], NAVSIM [15]. Notably, our approach achieves strong performance and significantly reduces the average collision rate compared to state-of-the-art methods on nuScenes. Additionally, using only camera inputs, we achieve PDMS performance comparable to state-of-the-art methods that utilize both camera and LiDAR data on the NAVSIM benchmark. These findings both enhance autonomous driving safety and underscore the potential of learning from video-based environmental representations.

The main contributions of this work can be summarized as follows:

- We propose Policy World Model, which unifies world modeling and trajectory planning, and more importantly, is able to benefit planning by unleashing the learned world knowledge through action-free future forecasting.

- We develop a parallel video forecasting approach that accelerates prediction while preserving visual coherence, supported by a context-guided tokenizer for compact representation and a dynamic focal loss for emphasizing temporally varying regions.

- Our method, relying solely on front camera input, performs on par with or even surpasses state-of-the-art multi-view and multi-modal driving approaches on widely used benchmarks, while achieving efficient visual generation results, supporting safe and efficient planning.

## 2 Related Work

**End-to-end Autonomous Driving.** End-to-end autonomous driving has advanced remarkably rapidly. Modern systems [16–19] map raw sensor inputs directly to control actions, thereby simplifying the traditional perception-to-control pipeline. Early methods [20, 21] used structured bird's-eye view (BEV) representations, while later work adopted dense 3D occupancy grids [22, 23] or sparse queries [24, 25] for richer, more efficient scene understanding, though handcrafted intermediate representations can limit generalization. More recently, autonomous driving research has begun to incorporate large language models (LLMs) [26, 27] and multimodal LLMs (MLLMs) [28–30]: driving scenes can be encoded as text for reasoning, achieving strong results in simulators [31–33], while video-based approaches leverage them to predict actions and answer queries [34–37].

**Generative World Models.** World models enable agents to understand and predict future environments from experience [38–44]. In autonomous driving, various approaches have been proposed for world model construction. Bevworld [45] and WoTE [46] operate in the BEV latent space, while Drive-OccWorld [10] predicts future occupancy states to interact with planning modules. For video generation, generative world models synthesize controllable and high-quality driving scenes, typically through large-scale pretraining. These models fall into two main categories: diffusion-based and autoregressive.Diffusion-based methods include DriveDreamer [4] for controllable video generation, Vista [8] for large-scale conditional synthesis, GLAD [47] for arbitrary-length videos, and DriveDreamer-2 [5] as well as Drive-WM [48] for multi-view generation. Autoregressive models such as GAIA-1/2 [7, 49] unify frames, text, and actions into a single sequence for multi-condition control and long-horizon prediction, while InfinityDrive [50] and DrivingWorld [51] extend to longer and continuous videos. However, token-by-token prediction makes these models computationally expensive and significantly less practical for efficiency-critical downstream tasks.

**Unified Generation and Understanding Models.** Building on the rapid progress of LLMs and MLLMs, recent research increasingly emphasizes unifying both visual understanding and generative capabilities within a single MLLM [52–54], often by carefully aligning the representations from large pretrained visual encoders with the embedding space of LLMs. Pioneering unified multimodal models (UMMs) [55–57] further extend this integration through powerful autoregressive or diffusion-based modeling paradigms. Such holistic unification provides a more principled and scalable framework to externalize the implicit world knowledge encoded in pretrained MLLMs, thereby significantly enhancing their potential for comprehensive world modeling and downstream reasoning.

## 3 Method

We present Policy World Model (PWM), a unified model that integrates both world modeling and trajectory planning under a cohesive framework. Given historical video frames, PWM is able to stimulate future states by generating plausible future video frames in an action-free manner. Serving as a policy model, PWM can leverage its learned world knowledge to explicitly benefit planning via a state-action collaborative prediction scheme. We implement PWM using an end-to-end Transformer with an image tokenizer (See Figure 2), where the tokenizer is responsible for encoding input frames into a compressed token space with context guidance, and the Transformer learns to jointly represent the world and perform planning through auto-regressive prediction. In the following, we present the model and training details of the tokenizer as well as PWM in Sec. 3.1 and Sce. 3.2, respectively.

### 3.1 Image Tokenizer with Context-Guided Compression and Decoding

Most driving world models [11, 7, 49] prioritize the quality of the video generation. In order to capture high-resolution details, they typically adopt a long sequence of tokens to represent a single image, leading to significant computational and memory overhead during autoregressive generation. In contrast, PWM focuses on using the learned world representation to benefit planning rather than pursuing photo-realistic details. To achieve a balance between generation quality and efficiency, we employ a new image tokenizer with context-guided compression and decoding.

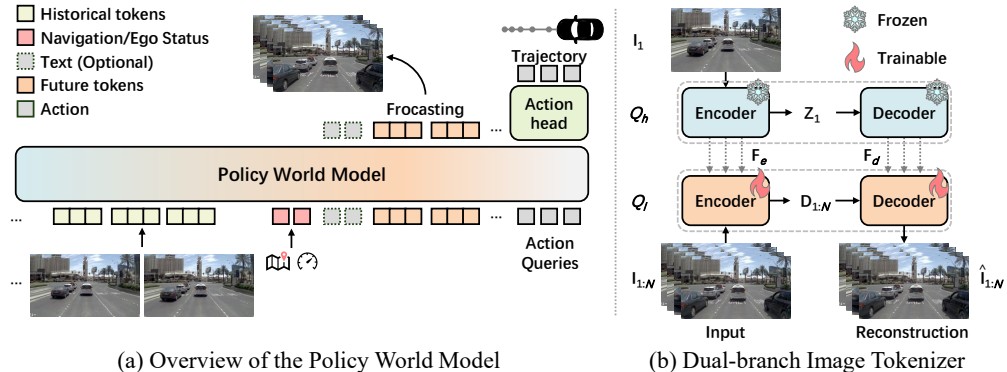

(a) Overview of the Policy World Model  (b) Dual-branch Image Tokenizer

Figure 2: **Policy World Model for Unified Forecasting and Planning.** (a) PWM leverages its pre-trained world modeling to generate future frames, enabling seamless collaboration between perception, prediction, and planning.(b) Future video frames $\mathbf{I}_{1:N}$ are compressed into compact latent representations guided by initial frame $\mathbf{I}_1$ and $\hat{\mathbf{I}}_{1:N}$ represents the reconstructed frames.

Given a video clip $\mathbf{I}_{1:N}$ containing $N$ frames, we aim to tokenize them into a compressed latent space by exploring their temporal consistency. To this end, we design the tokenizer as a two-branch encoder-decoder structure as depicted in Figure 2 (b). To capture high-fidelity visual content, the high-resolution branch $Q_h$ (blue blocks) takes as input the initial frame $\mathbf{I}_1$ with a full resolution. The encoder maps the initial frame into a sequence of $L$ tokens $\mathbf{Z}_1 = \{\mathbf{z}_1^1, \mathbf{z}_1^2, \ldots, \mathbf{z}_1^L\}$, which is further decoded back into the image space by the decoder. The encoder and decoder also produce a hierarchy of multi-scale intermediate feature maps $\mathbf{F}_e$ and $\mathbf{F}_d$, respectively, which are used as guidance to facilitate token compression. In-parallel to the high-resolution branch, the low-resolution branch $Q_l$ (orange blocks) will take all the downsampled frames in a batch as input and performs tokenization and reconstruction for each frame independently. For frame $\mathbf{I}_t$, a compressed token sequence $\mathbf{D}_t = \{\mathbf{d}_t^1, \mathbf{d}_t^2, \ldots, \mathbf{d}_t^{L'}\}$ will be generated by the encoder, with $L' << L$. To transfer guidance information $\mathbf{F}_e$ and $\mathbf{F}_d$ from the high-res to low-res branch, the encoder and decoder of the two branches are laterally connected via cross-attention layers. With the help of these guidance, the low-resolution branch is able to effectively represent and reconstruct the input frames using a significantly reduced number of image tokens.

In our experiment, we adopt a pre-trained image tokenizer [55] as the high-resolution branch $Q_h$, which is freezed during training. The low-resolution branch $Q_l$ implements a learnable adaptation network mirroring the structure of $Q_h$, augmented with additional random initialized cross-attention and downsampling blocks. The encoder of $Q_l$ is able to tokenize an input frame of $128 \times 224$ resolution into a compact feature map of $4 \times 7$, giving rise to $L' = 28$ tokens per frame. We follow the standard VQ-GAN optimization strategy [58] to train the tokenizer. More implementation details are provided in the supplementary material.

### 3.2 Policy World Model

Most existing driving world models either solely focus on world simulation [48, 6] or perform world representation and trajectory planning as two separate tasks with a single model [11–13]. In contrast, our PWM serves as a more cohesive paradigm, which not only integrates future state forecasting and planning within unified architecture, but also ensures the learned world modeling and forecasting ability can explicitly benefit trajectory planning via a collaborative state-action prediction scheme. This is achieved by the following two unique techniques.

**Learning World Modeling from Action-Free Video Generation.** Most previous autonomous driving methods learn the video world model via action-conditioned video generation, which is highly dependent on labeled training data. More importantly, they fail to perform any forecasting before the action is predicted. Therefore, when the world model is integrated with the policy model in a unified structure [11–13], their future forecasting ability cannot be fully explored to benefit planning. To circumvent this issue, we propose to pretrain PWM on action-free video generation (Figure 3 (a)). Formally, given a video clip $\mathbf{I}_{1:N}$, we first transform it into image tokens $\{\mathbf{Z}_1, \mathbf{D}_{1:N}\}$ using the

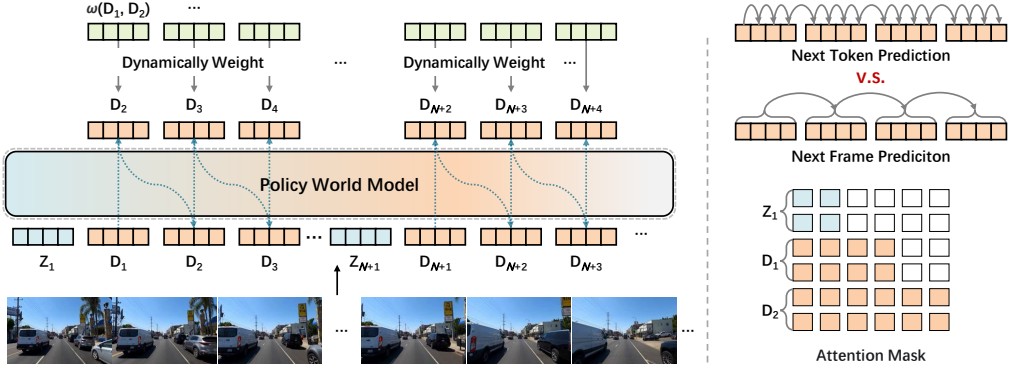

| (a) World Modeling via Action-free Video Generation | (b) Token Generation & Attention |

Figure 3: **Pipeline for video world modeling.** (a) World modeling is conducted on action-free, highly compressed video data using dynamically enhanced parallel prediction. (b) Comparison of token prediction formats and attention interactions.

tokenizer in Sec. 3.1, where $\mathbf{Z}_1 = \{\mathbf{z}_1^1, \mathbf{z}_1^2, \ldots, \mathbf{z}_1^L\}$ denotes the contextual tokens of the initial frame, and $\mathbf{D}_t = \{\mathbf{d}_1^1, \mathbf{d}_1^2, \ldots, \mathbf{d}_1^{L'}\}$ denote the compressed tokens of the $t$-th frame. PWM is then trained to generate these video token sequence in an autoregressive manner. Thanks to the token compression scheme, PWM can simultaneously generate all the tokens of a image in-parallel, allowing us to replace the conventional next-token generation with the more efficient next-frame generation scheme (See Figure 3 (b)):

$$P(\mathbf{D}_{1:N}; \mathbf{Z}_1) = \prod_{t=2}^{N} P_\theta(\mathbf{D}_t | \mathbf{D}_{<t}; \mathbf{Z}_1), \tag{1}$$

where $\theta$ denotes the PWM parameters. To better model the spatial relationship between tokens of the same frame, we employ the bi-directional attention within one frame, with the attention mask illustrated in Figure 3 (b).

During training, we empirically observed that a significant proportion (up to 50%) of the frame tokens are unchanged across adjacent frames, which makes the model tend to predict static tokens, impeding its temporal dynamic modeling ability. To alleviate this issue, we introduce a Dynamic Focal Loss (DFL) that emphasize temporally varying image regions through spatial weighting. Specifically, we define a dynamic weighting function $\omega(\mathbf{d}_t^i, \mathbf{d}_{t-1}^i)$ that measures the contribution of each token prediction based on its temporal change:

$$\omega(\mathbf{d}_t^i, \mathbf{d}_{t-1}^i) = \alpha \, \mathbb{I}\big[\mathbf{d}_t^i \neq \mathbf{d}_{t-1}^i\big] + \beta \, \mathbb{I}\big[\mathbf{d}_t^i = \mathbf{d}_{t-1}^i\big], \quad \alpha > \beta, \tag{2}$$

where $\mathbb{I}[\cdot]$ is the indicator function and $\alpha$, $\beta$ are hyperparameters that control the relative importance of dynamic versus static token predictions. The overall DFL for the $t$-th frame is then formulated as:

$$\mathcal{L}_{\text{DFL}} = -\sum_{i=1}^{L'} \omega(\mathbf{d}_t^i, \mathbf{d}_{t-1}^i) \log P_\theta\big(\mathbf{d}_t^i \mid \mathbf{D}_{<t}; \mathbf{Z}_1\big), \tag{3}$$

This design encourages the model to allocate more attention to dynamic regions, thereby enhancing its ability to capture meaningful spatio-temporal variations across frames.

**End-to-End Planning with Future State Forecasting.** In order to explicitly benefit planning with the attained world modeling ability, PWM is further fine-tuned for end-to-end planning in an autoregressive manner. To this end, each data sample for fine-tuning is prepared as a multimodal sequence $\{\mathbf{D}_{1:t}, \mathbf{E}_t, \mathbf{X}_t, \mathbf{D}_{t+1:t+n}, \mathbf{A}_{1:m}\}$. As shown in Figure 2, the inputs to PWM include the image tokens of observed frames $\mathbf{D}_{1:t}$ and the current navigation command with ego status $\mathbf{E}_t$. It then learns to autoregressively predict the textual tokens $\mathbf{X}_t$ describing the current scenario and plausible future frame tokens $\mathbf{D}_{t+1:t+n}$ using its pre-trained world modeling ability as Eq.(3). Subsequently, $m$ learnable action tokens are fed into PWM, and interact with the predicted latent features of the textual description and future forecasts. Their outputs are decoded into trajectory coordinates $\mathbf{A}_{1:m}$ by a lightweight action head. For training, $\mathbf{X}_t$ and $\mathbf{D}_{t+1:t+n}$ are supervised with cross-entropy loss, and

$\mathbf{A}_{1:m}$ with L1 loss. Through the above collaborative state-action prediction scheme, PWM is able to explicitly leverage the forecasted future states, yielding more reliable trajectory planning.

**Discussion of Collaborative State-action Prediction.** Our use of **collaborative** denotes a causal and knowledge-sharing relationship within PWM, leveraging learned world knowledge to better facilitate action prediction. This differs from prior methods that build world model $\mathbf{P}_w(\mathbf{D}_{t+1:t+n}|\mathbf{D}_{1:t})$ and policy model $\mathbf{P}_\theta(\mathbf{A}_{1:m}|\mathbf{D}_{1:t})$ seperately. This is also disinct from recent attempts that integrate both world modeling and planning in a unified architecture $\mathbf{P}_\theta(\cdot|\mathbf{D}_{1:t})$, where future frames and actions are still predicted independently via $\mathbf{P}_\theta(\mathbf{D}_{t+1:t+n}|\mathbf{D}_{1:t})$ and $\mathbf{P}_\theta(\mathbf{A}_{1:m}|\mathbf{D}_{1:t})$ in downstream tasks, and knowledge is not explicitly shared between the two processes. In comparison, our PWM not only integrates world modeling and planning in a unified model $\theta$, but also unleashes the learned world knowledge through the proposed planning with future state forecasting scheme which can be expressed as:

$$\mathbf{P}_\theta(\mathbf{D}_{t+1:t+n}|\mathbf{D}_{1:t}) \cdot \mathbf{P}_\theta(\mathbf{A}_{1:m}|\mathbf{D}_{1:t+n}) \rightarrow \mathbf{P}_\theta(\mathbf{D}_{t+1:t+n}\mathbf{A}_{1:m}|\mathbf{D}_{1:t}) \tag{4}$$

As shown in Eq.(4), world modeling and planning are not performed independently but in a **collaborative** manner, allowing the PWM to mimic the human-like anticipatory ability based on the learned world knowledge and perform more reliable planning.

## 4 Experiment

### 4.1 Experimental Setups

**Datasets.** We adopt OpenDV-YouTube dataset [3], nuScenes [14], and NAVSIM [15] as main training datasets. OpenDV-YouTube contains 1747 hours of front-camera video at 10 Hz from 244 cities with scene description generated by BLIP2 [59]. nuScenes includes 1000 scenes (5.5 hours in total) with six camera views per scene, and is annotated with six 3-second waypoints and text descriptions [60]. NAVSIM is built upon OpenScene [61], including 103k training and 12k test samples. We train the tokenizer on the the OpenDV-YouTube. The PWM model is pretrained on OpenDV-YouTube for action-free video generation and then finetuned on nuScenes and NAVSIM for planning. We use only front-view camera video, with 10 Hz for OpenDV-YouTube and NAVSIM, and 12 Hz for nuScenes.

**Metrics.** To assess the quality and consistency of future video generation, we employ three standard video synthesis metrics: FVD [62], LPIPS [63], and PSNR [64]. For the planning tasks, we use L2 error (m) and collision rate (%) on nuScenes, and Predictive Driver Model Score (PDMS) on NAVSIM. PDMS is a composite metric designed to better align with closed-loop behavior, which combines five sub-scores: no-at-fault collisions (NC), drivable area compliance (DAC), time-to-collision (TTC), comfort (Comf.), and ego progress (EP).

**Implementation Details.** Our model is initialized from Show-o [55]. During tokenizer training, the frozen branch processes $256 \times 448$ images to produce 448 tokens per frame, while the trainable branch uses $128 \times 224$ images, generating 28 tokens. Each branch has a separate codebook of size 8192. We sample non-overlapping clips from the OpenDV-YouTube dataset, using 1% for validation and the rest for training. Training runs for $8 \times 10^5$ steps using AdamW on 6 NVIDIA A800 GPUs with a learning rate of $2 \times 10^{-4}$. After training, the tokenizer is frozen. Next, we pre-train the autoregressive world model on OpenDV-YouTube for $3 \times 10^5$ steps using AdamW with a learning rate of $1 \times 10^{-4}$. To prevent catastrophic forgetting, we also conduct auxiliary training on CC12M [65] and FineWeb [66] for image captioning and text generation. For downstream tasks, we fine-tune the model using only the front-view camera. On nuScenes, we condition on 1 second of history to predict 11 frames and 6 waypoints over 3 seconds. Training runs for 16 epochs on 2 A800 GPUs with batch size 8 and learning rate $3 \times 10^{-5}$. On NAVSIM, we use 2 seconds of history to predict 10 frames and 8 waypoints across 4 seconds, training for 20 epochs with batch size 14. Dynamic Focal Loss is used throughout. No data augmentation is applied. More details are provided in the Appendix A.

### 4.2 Overall Comparison

On the nuScenes dataset, the relatively simple driving scenarios tend to induce an over-reliance on the vehicle's ego status [68, 74]. Therefore, in Table 1, we compare planning performance both without and with access to ego status across several popular methods. PWM achieves the lowest average collision rates (%) of 0.07 and 0.04 across two settings, respectively, surpassing previous

Table 1: **Comparison on the nuScenes validation split.** Metrics are computed following the same protocol as [67]. For a fair comparison, results are reported separately for settings without and with ego status (marked with "†"); results for UniAD and VAD are reproduced from BEV-Planner [68]. The best results are bolded.

| Method | L2(m)↓ | | | | Collision(%)↓ | | | |
|---|---|---|---|---|---|---|---|---|
| | 1s | 2s | 3s | Avg. | 1s | 2s | 3s | Avg. |
| ST-P3 [20] | 1.59 | 2.64 | 3.73 | 2.65 | 0.69 | 3.62 | 8.39 | 4.23 |
| UniAD [21] | 0.59 | 1.01 | 1.48 | 1.03 | 0.16 | 0.51 | 1.64 | 0.77 |
| VAD-Base [67] | 0.69 | 1.22 | 1.83 | 1.25 | 0.06 | 0.68 | 2.52 | 1.09 |
| BEV-Planner [68] | 0.30 | 0.52 | 0.83 | 0.55 | 0.10 | 0.37 | 1.30 | 0.59 |
| Omni-Q [60] | 1.15 | 1.96 | 2.84 | 1.98 | 0.80 | 3.12 | 7.46 | 3.79 |
| LAW [69] | **0.24** | 0.46 | 0.76 | 0.49 | 0.08 | 0.10 | 0.39 | 0.19 |
| Drive-OccWorld [10] | 0.25 | **0.44** | **0.72** | **0.47** | 0.03 | 0.08 | 0.22 | 0.11 |
| PWM(Ours) | 0.41 | 0.75 | 1.17 | 0.78 | **0.01** | **0.01** | **0.18** | **0.07** |
| UniAD† [21] | 0.20 | 0.42 | 0.75 | 0.46 | 0.02 | 0.25 | 0.84 | 0.37 |
| VAD-Base† [67] | 0.17 | 0.34 | 0.60 | 0.37 | 0.04 | 0.27 | 0.67 | 0.33 |
| BEV-Planner† [68] | 0.16 | 0.32 | 0.57 | 0.35 | **0.00** | 0.29 | 0.73 | 0.34 |
| OccWorld-D† [23] | 0.39 | 0.73 | 1.18 | 0.77 | 0.11 | 0.19 | 0.67 | 0.32 |
| Omni-Q† [60] | **0.14** | **0.29** | **0.55** | **0.33** | **0.00** | 0.13 | 0.78 | 0.30 |
| RDA-Driver† [70] | 0.17 | 0.37 | 0.69 | 0.41 | 0.01 | 0.05 | 0.26 | 0.11 |
| DiffusionDrive† [17] | 0.27 | 0.54 | 0.90 | 0.57 | 0.03 | 0.05 | 0.16 | 0.08 |
| PWM(Ours)† | 0.20 | 0.38 | 0.65 | 0.41 | 0.01 | **0.02** | **0.09** | **0.04** |

Table 2: **NAVSIM NavTest split comparison.** Overall Predictive Driver Model Score (PDMS) and sub-scores reflecting closed-loop performance. C: multi-view camera; SC: single-view camera; C&L: multi-view camera + LiDAR; "-": no visual input. The best results are bolded.

| Method | Input | NC↑ | DAC↑ | EP↑ | TTC↑ | Comf.↑ | PDMS↑ |
|---|---|---|---|---|---|---|---|
| Human | - | 100.0 | 100.0 | 87.5 | 100.0 | 99.9 | 94.8 |
| Constant Velocity | - | 69.9 | 58.8 | 49.3 | 49.3 | **100.0** | 21.6 |
| Ego Status MLP | - | 93.0 | 77.3 | 62.8 | 83.6 | **100.0** | 65.6 |
| VADv2 [18] | C&L | 97.2 | 89.1 | 76.0 | 91.6 | **100.0** | 80.9 |
| TransFuser [71] | C&L | 97.7 | 92.8 | 79.2 | 92.8 | **100.0** | 84.0 |
| DRAMA [72] | C&L | 98.0 | 93.1 | 80.1 | 94.8 | **100.0** | 85.5 |
| Hydra-MDP [19] | C&L | 98.3 | 96.0 | 78.7 | 94.6 | **100.0** | 86.5 |
| DiffusionDrive [17] | C&L | 98.2 | **96.2** | **82.2** | 94.7 | **100.0** | **88.1** |
| UniAD [21] | C | 97.8 | 91.9 | 78.8 | 92.9 | **100.0** | 83.4 |
| LTF [71] | C | 97.4 | 92.8 | 79.0 | 92.4 | **100.0** | 83.8 |
| PARA-Drive [73] | C | 97.9 | 92.4 | 79.3 | 93.0 | 99.8 | 84.0 |
| LAW [69] | C | 96.4 | 95.4 | 81.7 | 88.7 | 99.9 | 84.6 |
| DrivingGPT [11] | SC | **98.9** | 90.7 | 79.7 | 94.9 | 95.6 | 82.4 |
| PWM(Ours) | SC | 98.6 | 95.9 | 81.8 | **95.4** | **100.0** | **88.1** |

state-of-the-art models including Drive-OccWorld [10] and DiffusionDrive [17], with the former adopting a decoupled world modeling paradigm. On the more challenging NAVSIM dataset, as shown in Table 2, PWM delivers obvious superiority. Although we rely on only a single front camera view, our framework significantly outperforms all prior camera-based models, including world modeling methods such as DrivingGPT and LAW. It achieves a PDMS score of 88.1, comparable to the state-of-the-art method DiffusionDrive, which uses both camera and LiDAR inputs. Meanwhile, our model achieves a higher time-to-collision (TTC) score of 95.4 and a no-at-fault collision (NC) score of 98.6.

## 4.3 Ablation Study

**Impact of Action-free Video World Knowledge.** We investigate the influence of learning from large-scale videos on planning performance in Table 3a and 3b. The first rows of the two tables show the results without pre-training, where models are fine-tuned on the downstream task using the base model's weights [55]. All subsequent rows report the results after pre-training. It shows that without pre-training, the model struggles to capture and predict dynamic scene changes and yields

Table 3: **Impact of world modeling and dynamic focal loss on nuScenes and NAVSIM.** "Pre-train" indicates training on Open-Youtube video."Fine-tune" indicates training on downstreaming benchmarks. "$\mathcal{L}_{\text{DFL-p}}$" and "$\mathcal{L}_{\text{DFL-f}}$" indicate whether Dynamic Focal loss (DFL) was used in Pre-train and Fine-tune, respectively. The video metrics and planning scores are reported.

(a) **Ablation study on nuScenes Dataset.**

| Pre-train | $\mathcal{L}_{\text{DFL-p}}$ | $\mathcal{L}_{\text{DFL-f}}$ | Visual Forecast Quality | | | Planning Metrics | |
|---|---|---|---|---|---|---|---|
| | | | LPIPS↓ | PSNR↑ | FVD↓ | Avg.L2(m)↓ | Avg.Col(%)↓ |
| ✗ | ✗ | ✗ | 0.27 | 21.07 | 826.15 | 3.34 | 1.51 |
| ✓ | ✗ | ✗ | 0.24 | 22.16 | 239.13 | 2.29 | 1.05 |
| ✓ | ✗ | ✓ | 0.24 | 22.24 | 96.53 | 1.23 | 0.56 |
| ✓ | ✓ | ✗ | 0.22 | 22.88 | 96.99 | 1.04 | 0.26 |
| ✓ | ✓ | ✓ | 0.22 | 23.07 | 67.13 | 0.78 | 0.07 |

(b) **Ablation study on NAVSIM Dataset.**

| Pre-train | $\mathcal{L}_{\text{DFL-p}}$ | $\mathcal{L}_{\text{DFL-f}}$ | Visual Forecast Quality | | | Planning Metrics | | | | | |
|---|---|---|---|---|---|---|---|---|---|---|---|
| | | | LPIPS↓ | PSNR↑ | FVD↓ | NC↑ | DAC↑ | EP↑ | TTC↑ | Comf.↑ | PDMS↑ |
| ✗ | ✗ | ✗ | 0.27 | 19.9 | 431.47 | 97.3 | 89.7 | 69.1 | 92.2 | 100.0 | 77.8 |
| ✓ | ✗ | ✗ | 0.25 | 20.71 | 199.79 | 97.7 | 90.8 | 72.6 | 93.7 | 99.9 | 80.7 |
| ✓ | ✗ | ✓ | 0.24 | 21.06 | 114.85 | 98.3 | 94.5 | 73.4 | 95.1 | 100.0 | 83.5 |
| ✓ | ✓ | ✗ | 0.23 | 21.22 | 110.95 | 98.3 | 94.5 | 80.4 | 94.4 | 100.0 | 86.3 |
| ✓ | ✓ | ✓ | 0.23 | 21.57 | 85.95 | 98.6 | 95.9 | 81.8 | 95.4 | 100.0 | 88.1 |

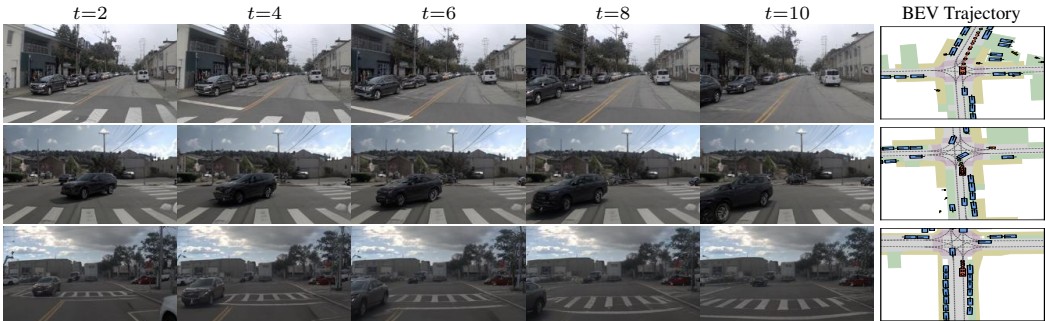

Figure 4: Decoded future-frame forecasting and corresponding BEV trajectory visualizations on NVASIM (green: GT, orange: prediction).

the worst prediction and planning metrics. After pre-training on action-free video generation, the model significantly improves its ability to predict future frames and achieves substantial gains in the planning task. In Table 5, we further compare the model's performance under different scales of video data on nuScenes. Without pre-training, adding the future-frame prediction task actually harms the model's planning capability, resulting in much worse performance than directly outputting trajectory waypoints. As the model is exposed to more data, from 0% to 50% and then to 100%, it progressively learns spatiotemporal modeling from driving videos, and the performance gap gradually narrows.

**Effectiveness of Dynamic Focal Loss.** We introduce a Dynamic Focal Loss to enhance the model's capability in capturing temporal dynamics for both video-frame prediction and planning. Tables 3a and 3b investigate the impact of applying this loss on prediction and planning performance for the nuScenes and NAVSIM datasets. Compared to omitting dynamic weight in both stages, applying it in either pre-training or fine-tuning yields clear improvements in three generation metrics as well as planning metrics. Notably, only using it during pre-training achieves stronger results on LPIPS and PSNR than fine-tuning alone, indicating more effective spatiotemporal modeling from large-scale video. This also translates into improved planning. Applying the loss in both stages yields the strongest improvements in video prediction and planning, confirming that bolstering temporal dynamics synergistically enhances both capabilities. Additional evaluation metrics on the OpenDV-YouTube are reported in Table 6. In Figure 4, we provide a qualitative visualization of the decoded future-frame predictions alongside the planned trajectories, illustrating their clear alignment.

Table 4: **Ablation study on visual forcasting on nuScenes and NAVSIM benchmark.**

| Forecast Horizon (Frames) | nuScenes | | | NAVSIM | | | | | | |
|---|---|---|---|---|---|---|---|---|---|---|
| | Avg.L2(m)↓ | Avg.Col(%)↓ | Latency | NC↑ | DAC↑ | EP↑ | TTC↑ | Comf.↑ | PDMS↑ | Latency |
| 0 | 0.80 | 0.13 | 0.88s | 98.0 | 95.1 | 82.4 | 94.1 | 99.9 | 87.3 | 0.57s |
| 5 | 0.82 | 0.10 | 1.01(+0.13)s | 98.4 | 95.4 | 81.5 | 94.8 | 100.0 | 87.7 | 0.69(+0.12)s |
| 10 | 0.78 | 0.07 | 1.13(+0.25)s | 98.6 | 95.9 | 81.8 | 95.4 | 100.0 | 88.1 | 0.85(+0.28)s |
| 15 | 0.80 | 0.09 | 1.26(+0.38)s | 98.7 | 95.8 | 81.4 | 95.4 | 100.0 | 88.0 | 0.97(+0.40)s |

Table 5: **Impact of different data scales of pre-training on nuScenes benchmark.**

| Data Usage | Forecasted Frames = 10 | | Forecasted Frames = 0 | |
|---|---|---|---|---|
| | Avg.L2(m)↓ | Avg.Col(%)↓ | Avg.L2(m)↓ | Avg.Col(%)↓ |
| 0% | 2.95 | 1.26 | 1.62 | 0.90 |
| 50% | 0.85 | 0.14 | 0.88 | 0.21 |
| 100% | 0.78 | 0.07 | 0.80 | 0.13 |

Table 6: **Effect of dynamic focal loss on pre-training.**

| $\mathcal{L}_{\text{DFL-p}}$ | Visual Forecast Quality | | |
|---|---|---|---|
| | LPIPS↓ | PSNR↑ | FVD↓ |
| ✗ | 0.23 | 20.29 | 211.26 |
| ✓ | 0.22 | 20.32 | 118.07 |

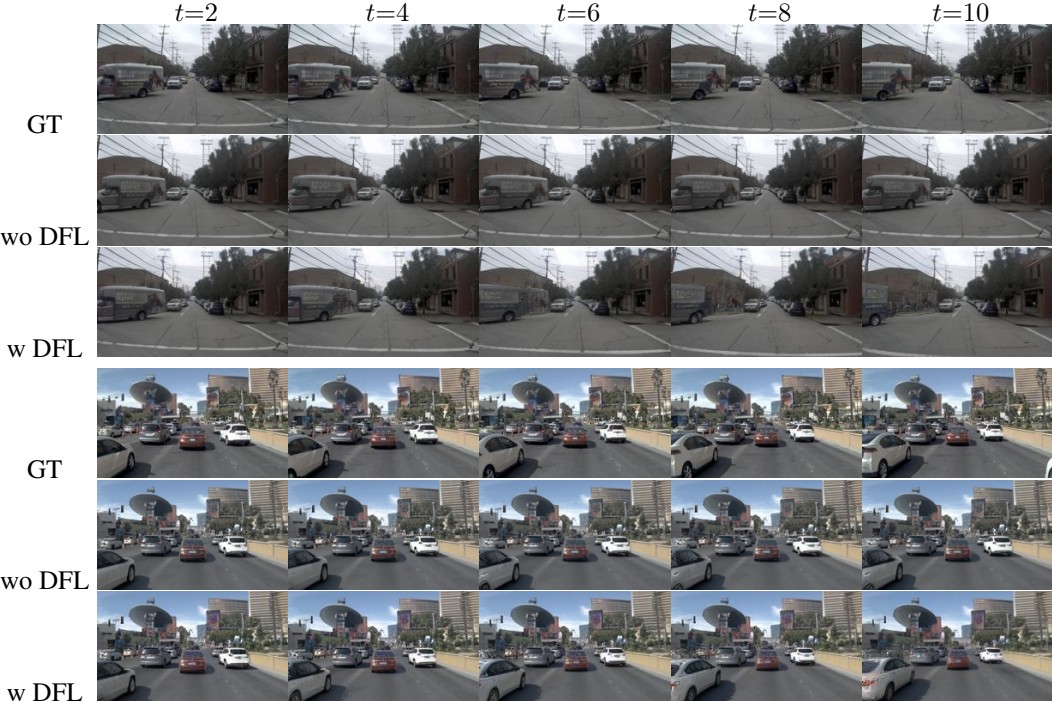

Figure 5: **Visualzation.** Comparison of future frame predictions with and without Dynamic Focal Loss (DFL). The first row shows ground truth frames, the second row shows predictions without DFL, and the third row shows predictions with DFL. Sampled frames at $t=2, 4, 6, 8, 10$ are shown.

To further illustrate the impact of DFL, we also provide a visual comparison in Figure 5. Each row corresponds to a specific model configuration: the first row shows ground truth future frames, the second row presents prediction results without DFL, and the third row shows results with DFL. These visualizations demonstrate that the use of DFL helps the model better capture and represent dynamic scene elements over time, resulting in more accurate and temporally coherent predictions.

**Influence of Video Forcasting on Planning.** We evaluate the impact of forecasting and efficiency on downstream planning performance under different number of predicted future frames in Table 4. Predicting 10 future frames achieves the best performance across both datasets. We speculate that shorter horizons capture insufficient temporal dynamics, resulting in weaker planning. In contrast, longer horizons degrade prediction quality and may introduce hallucinations, especially given the limited perception from a single front-view camera, ultimately impairing decision-making. We evaluate frame token forecasting efficiency on a single NVIDIA A800 GPU (batch size 1). As shown in Table 4, the additional latency over the zero-horizon baseline is marginal, and the model achieves about 40 FPS without pixel-space decoding.

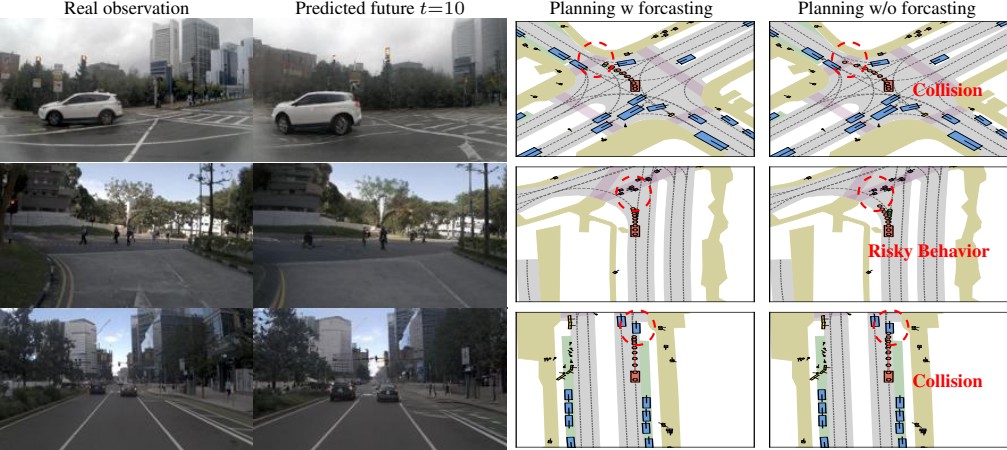

| Real observation | Predicted future $t{=}10$ | Planning w forcasting | Planning w/o forcasting |

Figure 6: Comparison of planning results with and without incorporating future prediction during training (green: GT, orange: prediction).

On the nuScenes benchmark, introducing future-frame prediction yields a substantial reduction in average collision rate. On the more challenging NAVSIM dataset, we observe a complementary trade-off: (i) when the model is trained without future prediction, it achieves a higher EP score, indicating that its planned trajectories advance farther along the route within the allotted horizon. By contrast, (ii) when the model does predict future frames during fine-tuning, it attains higher NC and TTC scores, demonstrating more effective avoidance of potential collisions, and a higher DAC score, showing that its trajectories remain better confined to drivable areas. Furthermore, in Figure 6 we present a qualitative analysis of how future-frame prediction affects planning on challenging NAVSIM. The first column shows the current observation, the second shows the model's tenth-frame prediction under forecasting fine-tuning, and the third and fourth columns respectively overlay the planned trajectories from the "with" and "without" prediction variants. From these findings, we infer that future-frame forcasting can induce a more conservative planning strategy, sacrificing some progress (EP) in order to secure higher safety margins (NC and TTC). As a result, the model favors lower-risk routes rather than maximizing forward progress.

## 5 Conclusion

In this paper, we propose the Policy World Model (PWM), a unified framework that integrates world modeling and trajectory planning for autonomous driving. By introducing action-free video generation and multi-modal reasoning, PWM can forecast future scenes and make informed decisions without relying on action-labeled data. Our design significantly improves planning performance and efficiency, achieving competitive results using only monocular camera input. This work highlights the potential of using compact, anticipatory video-based representations to drive safer and more scalable autonomous systems.

**Limitations and Future Work.** Although video-based PWM demonstrates strong performance, relying solely on single-view inputs can compromise robustness under poor visibility conditions. Furthermore, its short planning horizon limits its applicability in long-horizon scenarios. In future work, we aim to further explore efficient integration of multi-view inputs and enhance long-term forecasting capabilities to improve generalization and real-world readiness.

## 6 Acknowledgement

This paper is supported by the National Natural Science Foundation of China (62422610, U23A20386, 62441231, 62293542, 62276045, 62576072), Liao Ning Science and Technology Plan No.2023JH26/10200016 , Dalian Science and Technology Innovation Fund (2023JJ11CG001), Natural Science Foundation of Zhejiang (LD25F020001), Ningbo Key R&D project (2025Z039), and Ningbo Science and Technology Innovation Project (2024Z294).

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

# Supplementary Material Overview

This supplementary material provides additional implementation details, experimental setups, and extended results to complement the main paper.

- Section A describes architecture and training details of our image tokenizer and policy world model, including both pre-training and fine-tuning stages.
- Section B presents additional experiments on the nuScenes.
- Section C provides visualizations of the latent predictions and qualitative comparisons.

# A  Implementation and Experimental Details

## A.1  Image Tokenizer

**Architecture details.** We adopt a dual-branch architecture to map input images of different resolutions into a shared latent space. The trainable branch, denoted as $Q_l$, is initialized with the same structure as the frozen high-resolution branch $Q_h$. To enhance its modeling capacity, we insert additional self-attention layers after the multi-level downsampling stages in both the encoder and decoder. These layers are designed to better capture spatial relationships within the feature maps. Furthermore, we incorporate cross-attention layers to enable $Q_h$ to guide the learning of $Q_l$. This alleviates the burden on $Q_l$ to extract contextual information, allowing it to focus more on modeling temporal variations and dynamic changes. Specifically, multi-scale features from $Q_l$ are used as queries, while the corresponding features from $Q_h$ serve as keys and values. The attention is applied independently across scales. In the encoder of $Q_l$, self-attention and cross-attention is applied at resolutions of $8 \times 14$; in the decoder of $Q_l$, it is applied at $8 \times 14$, $16 \times 28$. Additionally, we introduce a lightweight MLP layer to perform $4\times$ downsampling and upsampling of latent token sequence length from $Q_l$ in the encoder and decoder, respectively. This serves to further compress and reconstruct the latent representations efficiently.

**More training details.** We sample the original Open-Youtube dataset into non-overlapping clips, each containing 40 frames spanning 4 seconds. During training, each sample randomly selects a continuous 30-frame segment, from which 2 frames are randomly chosen as high-resolution context frames. Subsequently, 8 future frames are randomly sampled as low-resolution video inputs. We optimize the model using the AdamW optimizer with 500 warm-up steps. Image reconstruction is supervised using the L1 loss. Since pixel-wise differences between future frames and initial context frames tend to increase over time, we apply a time-dependent weighting to the reconstruction loss, assigning greater emphasis to later frames to reflect their higher prediction difficulty and encourage better long-term modeling. We assign a weight of 2.0 to the perceptual loss and a weight of 1.0 to the discriminator loss to balance perceptual quality and realism during optimization.

## A.2  Policy World Model

### A.2.1  Pre-training setup

Although we sample two consecutive high-resolution initial frames for the tokenizer, only one high-resolution frame is actually fed into the model per second without supervision. This strategy does not significantly affect the model performance, while effectively reducing the input token sequence length and improving training efficiency. Each video clip is randomly cropped into a 24-frame continuous segment from the original 40-frame clip to define the prediction horizon. As shown in Figure 1(a), we mark the beginning and end of each high-resolution frame sequence using two special tokens, "<|soil|>" and "<|eoil|>", following the Show-o convention. For the compressed low-resolution frames, we introduce two additional special tokens, "<|sodl|>" and "<|eodl|>", to indicate the start and end positions of the low-resolution frames.

During training, the video autoregressive prediction task serves as the primary objective, while image-text captioning and pure language modeling tasks are incorporated to preserve the model's capability in understanding and generating language. These three tasks are mixed in a batch ratio of 3:1:1, respectively. The loss weights are set to 1.0 for the video task and 0.5 for both the captioning and text-only tasks. The warm-up steps are set to 1000.

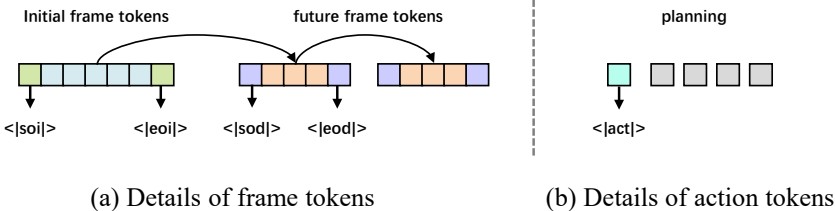

(a) Details of frame tokens      (b) Details of action tokens

Figure 1: **Detailed structure of input sequences.** (a) illustrates the format of high-resolution and low-resolution video frames. (b) depicts the input configuration used for trajectory prediction.

### A.2.2 Fine-tuning setup.

**Details on nuScenes.** We consider two configurations to investigate the influence of ego status on planning performance. Structurally, when ego status is included as input, we use a two-layer MLP with SiLU activation to project the ego state into the model's latent space. For navigation commands, we randomly initialize three learnable embeddings to represent "Go Straight," "Turn Left," and "Turn Right." The "<|act|>" token is used to prompt trajectory prediction, as illustrated in Figure 1(b). The action head is implemented as an MLP with SiLU activation. Our framework is designed to seamlessly support multi-modal outputs, including both video and language. Given that nuScenes is one of the most widely used open-loop datasets and that several prior works have proposed textual annotations based on it, we adopt a simplified setting where the action description serves as the target for language generation, using fixed prompts. For example: "*In this quiet nighttime driving situation, the vehicle should maintain a moderate speed and continue straight, adhering to lane discipline.*" This setup demonstrates the flexibility of our framework and lays the groundwork for future research to explore more expressive and diverse multi-modal outputs.

During training and evaluation, the last one second of each scene lacks future frames, which makes it unsuitable for future frame prediction. Therefore, in the training set, we manually exclude these final segments to ensure supervision consistency. For the validation set, we also discard the last one second of each scene when evaluating video generation metrics. However, to ensure a fair comparison with previous works on planning, we retain these segments when computing planning-related metrics.

**Details on NAVSIM.** We follow the official practice by concatenating the ego status and navigation command into a single vector, which is then projected into the model space via an MLP layer. Since this dataset does not provide textual descriptions, we do not include any language modeling tasks during training or inference. Additionally, some video frames required for prediction are not included in the NAVSIM dataset. To address this, we resample the missing frames from the original nuPlan dataset. Other aspects of the training procedure remain largely consistent with that of nuScenes.

## B    More experiments

### B.1    Set up in Dynamic Focal Loss

To effectively improve the video modeling and generation capability of the Policy World Model, we propose a Dynamic Focal Loss (DFL) that emphasizes temporally varying image regions through spatial weighting. We conduct an ablation study on the key hyperparameters $\alpha$ and $\beta$ on nuScenes, where $\alpha$ controls the weight for spatial tokens that change over time, and $\beta$ controls the weight for those that remain unchanged across consecutive frames. As shown in Table 1, we fix $\alpha$ to 1.0 and vary $\beta$ to explore their relative influence.

We observe that when $\alpha < \beta$ and $\beta$ is set to a small value (e.g., $\beta = 0.1$), the large disparity in task weights leads to overfitting in the future frame prediction task during training, while the planning task has not yet fully converged. This imbalance ultimately results in degraded overall performance, indicating the need to better coordinate the training progress of the two tasks for more effective joint optimization. On the other hand, when $\alpha \leq \beta$, the Dynamic Focal Loss mechanism tends to fail, leading to a significant drop in the quality of future frame prediction, which in turn negatively impacts the performance of the downstream planning task.

Table 1: **Ablation study on the hyperparameters in the Dynamic Focal Loss.** The dynamic weight $\alpha$ is fixed to 1.0 across all experimental settings.

| $\beta$ | Visual Forecast Quality | | | Planning Metrics | |
|---|---|---|---|---|---|
| | LPIPS↓ | PSNR↑ | FVD↓ | Avg.L2(m)↓ | Avg.Col(%)↓ |
| 0.1 | 0.23 | 22.69 | 65.07 | 0.82 | 0.12 |
| 0.4 | 0.22 | 23.07 | 67.13 | 0.78 | 0.07 |
| 0.7 | 0.22 | 23.11 | 71.84 | 0.84 | 0.09 |
| 1.0 | 0.22 | 22.88 | 96.99 | 1.04 | 0.26 |
| 2.0 | 0.22 | 22.65 | 93.83 | 1.27 | 0.27 |

Table 2: **Impact of the Textual Generation Task on nuScenes validation split.** We also conduct an ablation study to evaluate the impact of the textual generation task on planning performance. Specifically, we compare different settings: (1) "None": without incorporating any text generation task, (2)"Scene": with scene description prediction, and (3) "Action": with action description prediction.

| Text Task Type | Visual Forecast Quality | | | Planning Metrics | |
|---|---|---|---|---|---|
| | LPIPS↓ | PSNR↑ | FVD↓ | Avg.L2(m)↓ | Avg.Col(%)↓ |
| None | 0.22 | 23.12 | 65.45 | 0.77 | 0.09 |
| Scene | 0.22 | 22.97 | 67.52 | 0.77 | 0.08 |
| Action | 0.22 | 23.07 | 67.13 | 0.78 | 0.07 |

## B.2  Visualization and Analysis of Temporal Representations in Predicted Video Frames.

As shown in Figure 2, we visualize the 2D UMAP projection of forecasted latent video frames over time on the nuScenes validation split. Specifically, we extract future predicted driving latents from each 20-second-long scene and concatenate all samples across scenes. Each predicted frame is then projected into a 2D coordinate point according to its temporal order. Different colors are used to indicate the temporal progression from 0s to 20s.

We observe that, although the future representations are generated independently for each sample and conditioned on different input frames, the resulting projected embeddings consistently exhibit smooth and coherent temporal dynamics across the full prediction horizon. This phenomenon suggests that our Policy World Model is capable of learning a robust internal representation of the temporal evolution of driving scenes. Importantly, it also demonstrates that the model can effectively decouple the dynamics from specific visual content in the input, capturing underlying motion patterns that are consistent and semantically meaningful, regardless of the particular observation used as guidance.

## B.3  Impact of the Textual Generation Task on Planning Performance.

To isolate the impact of different types of text prediction on planning performance of nuScenes, we design three settings. The first setting excludes any text prediction task, which also serves as the baseline in the NAVSIM dataset. The second setting provides a detailed description of the scene environment, while the third offers a brief prediction of the ego vehicle's future behavior. For action descriptions, we filter out information related to multi-view perspectives and retain only a brief summary for supervision. As shown in Table 2, In our model, incorporating scene- or action-level textual generation tasks does not lead to significant improvements in future frame forecasting or downstream planning metrics, suggesting a limited effect in our specific setting.

# C  Visualizations

## C.1  Comparison visualization

In Figures 3 and 4, we provide additional qualitative results to compare planning with and without future frame prediction. On the left, we show a sequence of video frames where the red-bordered frame indicates the current observation, and the unframed ones are predicted future frames. We

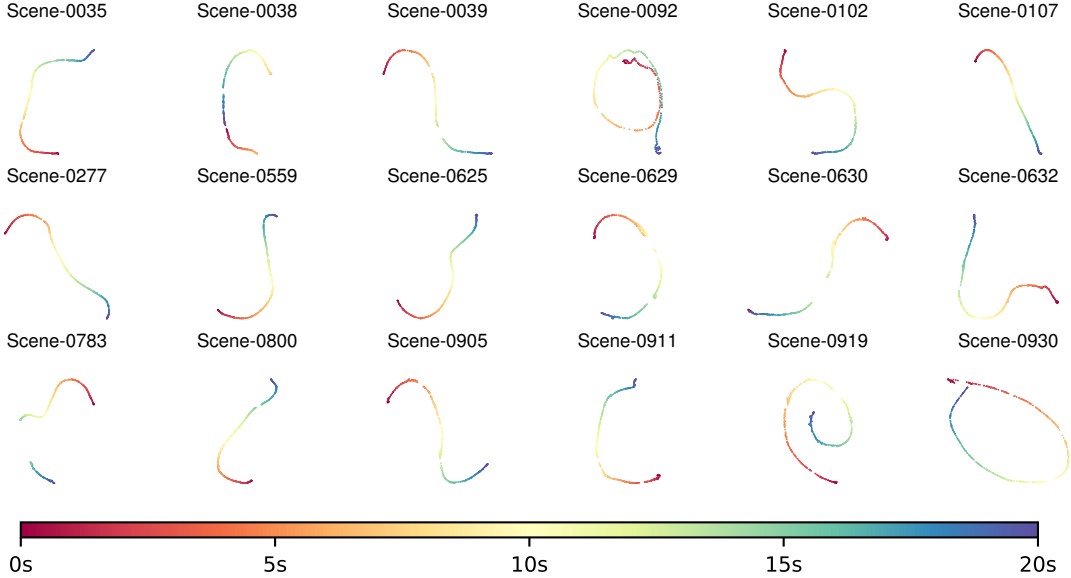

Figure 2: The UMAP projection reveals temporal changes in frame-level latent representations, highlighting the model's ability to capture scene dynamics across different environments.

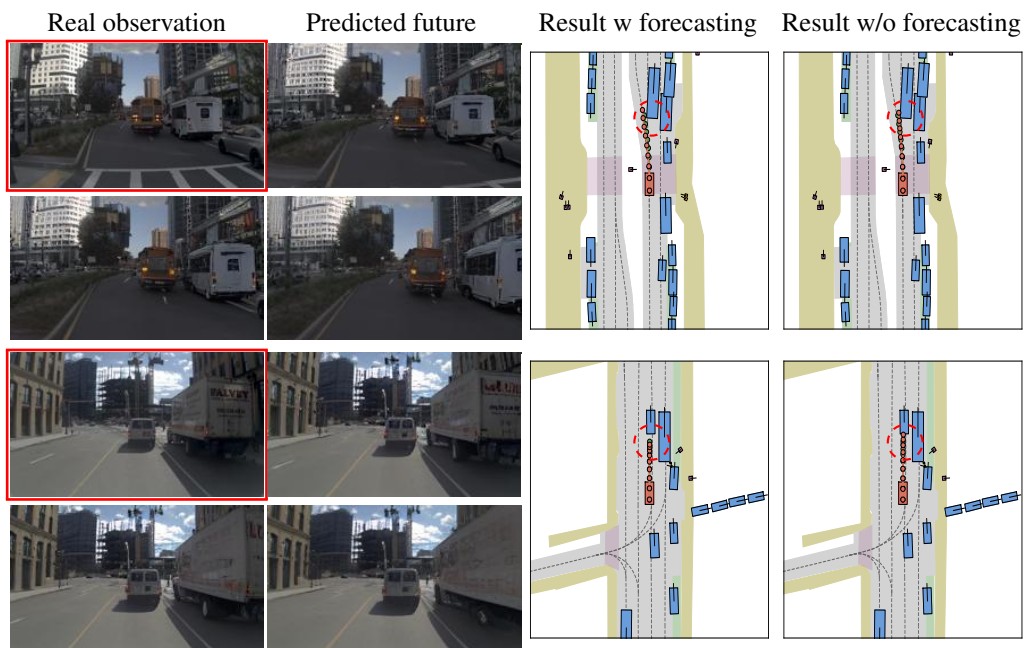

Figure 3: More comparison of planning results with and without incorporating future prediction during training (green: GT, orange: prediction).

uniformly sample three future frames for visualization. On the right, we present the corresponding BEV (bird's-eye view) planning outcomes. These comparisons highlight how incorporating future frame prediction can enhance planning quality by enabling better anticipation of dynamic scene changes. In Figure 5, we show additional visual comparisons illustrating the effect of Dynamic Focal Loss on future frame prediction.

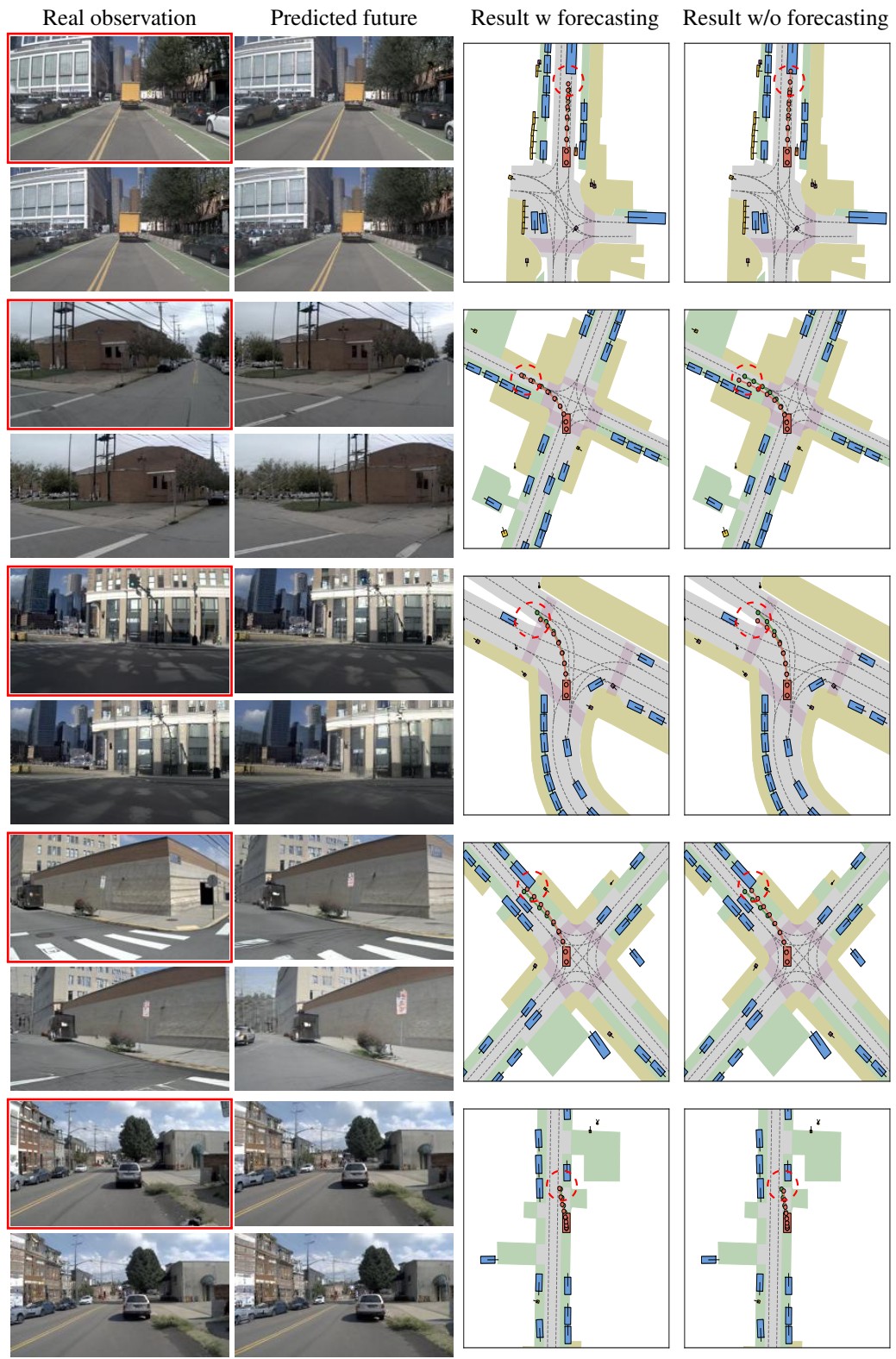

Figure 4: **Visualzation.** More comparison of planning results with and without incorporating future prediction during training (green: GT, orange: prediction).

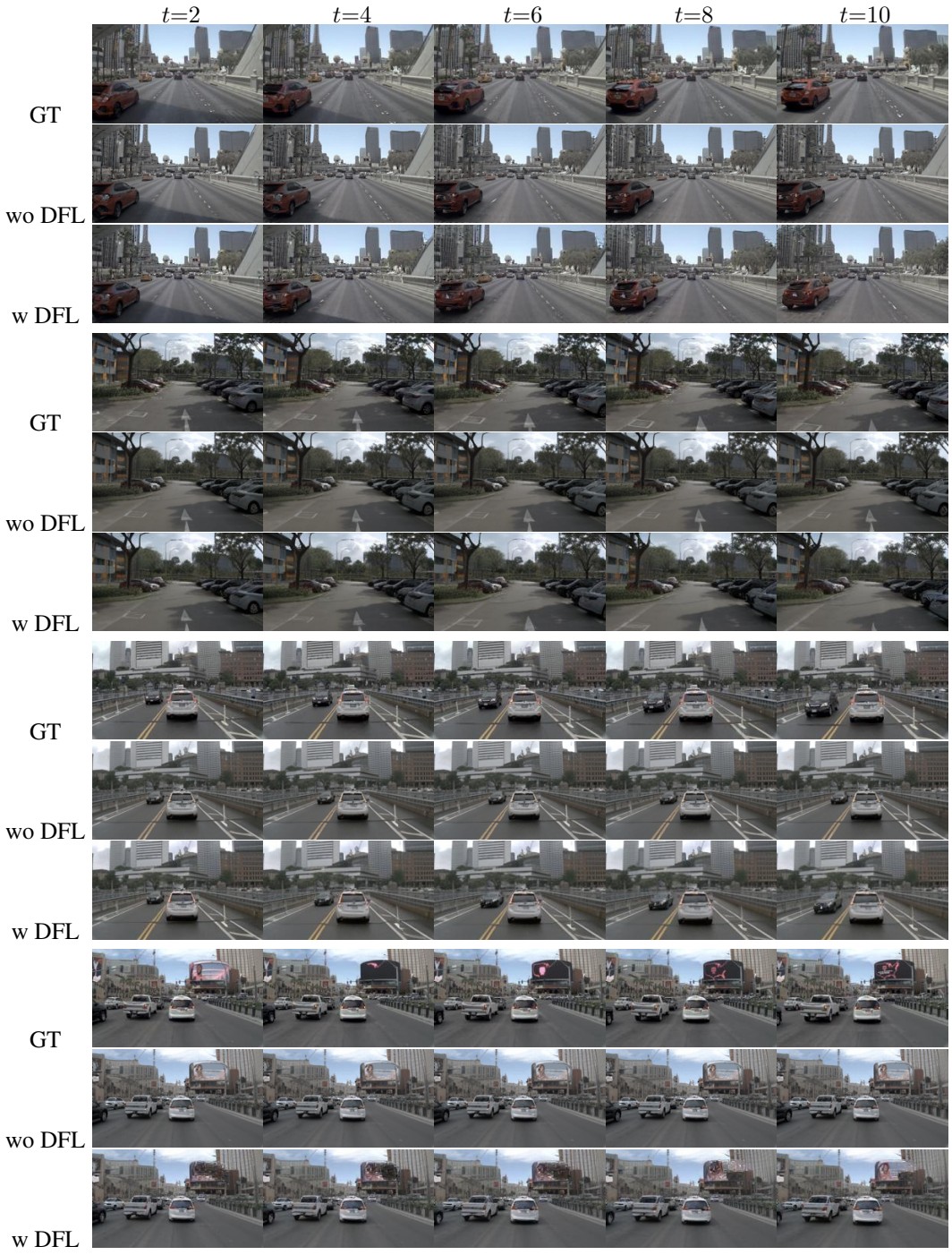

Figure 5: **Visualzation.** More comparison of future frame predictions with and without Dynamic Focal Loss (DFL). The first row shows ground truth frames, the second row shows predictions without DFL, and the third row shows predictions with DFL. Sampled frames at $t=2, 4, 6, 8, 10$ are shown.

