# OpenReview forum: "From Forecasting to Planning: Policy World Model for Collaborative State-Action Prediction"
_NeurIPS.cc/2025/Conference — NeurIPS 2025 poster_

### Official Review · Reviewer_NcJp · 2025-06-11

**Clarity:** 3
**Significance:** 3
**Originality:** 3
**Rating:** 5
**Confidence:** 5

**Summary:**

This paper proposes PWM (Policy World Model), a unified architecture that integrates world modeling and trajectory planning. By tightly coupling the world model with the planner, PWM enables the world model to directly benefit the planning process. To enhance efficiency, the paper also introduces a novel image tokenizer. The proposed method demonstrates strong performance on the NAVSIM and nuScenes benchmarks.

**Questions:**

1. As shown in Tables 3 and 4, the performance gain from pretraining (77.8 → 86.3) is more substantial than that from forecasting (87.3 → 88.1). This suggests that the key factor may not lie in the forecasting capability of the world model itself, but rather in the effectiveness of video pretraining in a world model-like framework.
2. Empirically, autoregressive learning tends to perform better when the signal-to-noise ratio is higher. In this work, 28 tokens are used to encode the future frames. Have you explored how varying the number of tokens affects the performance?

**Ethical Concerns:**

["NO or VERY MINOR ethics concerns only"]

**Final Justification:**

This paper presents PWM, which achieves performance comparable to existing BEV-based methods within an autoregressive framework on the NAVSIM benchmark, demonstrating its potential. The paper also investigates the impact of visual prediction quality on policy learning, offering meaningful insights. I recommend accepting this paper.

**Limitations:**

yes

**Quality:**

3

**Strengths And Weaknesses:**

**Strength:**

1. The paper is clearly presented, with a straightforward and compelling motivation.
2. Table 3 reveals a positive correlation between visual quality and planning performance, which is an interesting and noteworthy observation.
3. PWM achieves performance comparable to existing BEV-based methods within an autoregressive framework on the NAVSIM benchmark, highlighting its potential.
4. The proposed dual-branch tokenizer proves to be highly effective for video compression, significantly reducing the number of tokens.

**Weakness:**

1. The main concerns lie in computational efficiency and the lack of support for multi-sensor inputs.

---

> ### Author Rebuttal · Authors · 2025-07-31
>
> Thank you for your insightful review and recognition of PWM's strengths and contributions. Detaield responses are as follows.
>
> **W1: Concerns of computational efficiency and inadequate multi-sensor input support.**
>
> Efficiency:  In our response to Reviewer n8oM's comment (W3), we compare the inference costs between state-of-the-art autonomous driving methods. Our inference latency is comparable with MLLM-based competitors with leading driving performance. Detailed results will be added to the final version of our paper.
>
> Multi-sensor / multi-view support: While our current work does not explicitly focus on multi-view or multi-sensor fusion, we consider the autoregressive framework inherently extensible and well-suited for integrating multiple modalities. Although this is not the primary focus of our current study, we regard it as a promising direction for future research. Notably, concurrent works such as Gaia-2 have demonstrated the feasibility of multi-view autoregressive generation, which we find both insightful and inspiring for extending our method.
>
> **Q1: Pretraining vs. Forecasting in Performance Gain.**
>
> To better disentangle the effects of video pretraining and future frame forecasting, we conduct additional ablation studies on the NAVSIM dataset (containing 100k training samples). As shown in the following table, pre-training is indeed crucial for our model to learn world modeling capabilities. With our proposed forecasting mechanism, we can observe an additional driving performance gain, which indicates that the contributions of forcasting and pre-training are orthogonal to the final performance.
>
>
> | Pre-train | Forecast  | NC ↑        | DAC ↑       | EP ↑        | TTC ↑       | Comf. ↑     | PDMS ↑      |
> |:---------:|:--------:|:-----------:|:-----------:|:-----------:|:-----------:|:-----------:|:-----------:|
> | ✗         | ✗        | 97.3        | 94.5        | 81.4        | 92.1        | 100.0       | 85.6        |
> | ✓         | ✗     | 98.0 (+0.7) | 95.1 (+0.6) | 82.4 (+1.0) | 94.1 (+2.0) | 99.9 (-0.1) | 87.3 (+1.7) |
> | ✓         | ✓       | 98.6 (+1.3) | 95.9 (+1.4) | 81.8 (+0.4) | 95.4 (+3.4) | 100.0 (+0.0)| 88.1 (+2.5) |
>
> **Q2: The choice of token quantity and impact on performance.**
>
> In our implementation, the tokenizer used in the baseline model encodes a 128×224 image into a compressed feature map of shape 8×14, resulting in 112 tokens. In our work, we adopt a more compact representation by further compressing the spatial dimensions of the future frames, resulting in a 4×7 grid of tokens (i.e., 28 tokens) per frame.
>
> We acknowledge that exploring the effect of token count on downstream planning performance is a meaningful direction. However, such an investigation requires retraining the tokenizer, followed by full-scale pretraining and finetuning for each token configuration, which is computationally expensive. As a result, we have not yet completed this ablation.
>
> However, we note that prior work [1] has studied the effect of token count in visual tokenizers designed for generation tasks. An interesting observation as shown in Figure 4 of [1] is that increasing the number of tokens improves generation quality. When these tokens are directly used for perceptual tasks (e.g. ImageNet-1K linear probing) the performance tends to degrade. This suggests a potential trade-off: using more tokens helps capture texture-level information, while using fewer tokens yields more compact and semantically concentrated representations, which may be more beneficial for high-level tasks such as planning.
>
>
> [1]  Yu Q, Weber M, Deng X, et al. An image is worth 32 tokens for reconstruction and generation[C]. Advances in Neural Information Processing Systems, 2024, 37: 128940-128966.

---

> > ### Comment · Reviewer_NcJp · 2025-08-04
> >
> > After reading the rebuttal, I hope the authors will explore token compression and multi-sensor settings in future work. I was a bit confused by the first row of Table Q1 — since the point accuracy reported in the paper is 77.8, could you clarify the specific setting used here?

---

> > > ### Author Response · Authors · 2025-08-04
> > > **Thank you for the response**
> > >
> > > Dear Reviewer,
> > >
> > > We sincerely thank you for the time and effort you have dedicated to reviewing our paper as a volunteer and really appreciate your suggestions on the future work. The PDMS@77.8 is achieved via the baseline method that directly performs forcasting based planning without pre-training. Its performance is even lower than the baseline method (PDMS@85.6) that performs neither pre-trainng nor forcasting. This is reasonable since it has not learned the strong world knowledge from pre-training, the forcasting results are unreliable (LPIPS 0.27, PSNR 19.9, FVD 431.47 in the first row of Table 3(b) in our paper), leading to infereior planning performance. We are sorry for the confusion. Pleased refer to the following table for a more complete ablation study. We will also add detailed results as well as the dicussions in the revised paper.
> > >
> > >
> > > || &emsp; Pre-train | &emsp; Forecast &emsp; |    NC↑   &emsp;  | DAC↑   &emsp; | EP↑   &emsp;  | TTC↑  &emsp; | Comf.↑  &emsp;| **PDMS↑** &emsp; |
> > > |:-----:|:------------------:|:-----------------:|:--------------------:|:-------:|:-------:|:-------:|:-------:|:-------:|
> > > |1| ×         | ×       |  97.3                                | 94.5    | 81.4    | 92.1    | 100.0   | 85.6    |
> > > |2| ×         | √       |  97.3                                | 89.7    | 69.1    | 92.2    | 100.0   | 77.8 (-7.8)    |
> > > |3| √         | ×        | 98.0                                | 95.1    | 82.4    | 94.1    | 99.9    | 87.3  (+1.7)  |
> > > |4| √         | √        | 98.6                                | 95.9    | 81.8    | 95.4    | 100.0   | 88.1  (+2.5)  |

---

> > > > ### Comment · Reviewer_NcJp · 2025-08-05
> > > >
> > > > This table provides a clearer understanding of the crucial role that future prediction quality plays in policy learning. I will maintain my recommendation as Accept.

---

> > > > > ### Author Response · Authors · 2025-08-05
> > > > > **Thank you**
> > > > >
> > > > > Dear Reviewer,
> > > > >
> > > > > We are grateful for your positive assessment of our revised table, which helps clarify the crucial role of future prediction quality in policy learning. Your suggestion to explore token compression and multi-sensor settings in future work is highly valuable, and we fully intend to prioritize these directions to enhance the practicality and robustness of our method.
> > > > >
> > > > > In addition, we will carefully incorporate all insights from your review to further polish the manuscript.
> > > > >
> > > > > Wish you have a nice day!

---

### Official Review · Reviewer_WR4K · 2025-06-24

**Clarity:** 2
**Significance:** 3
**Originality:** 3
**Rating:** 4
**Confidence:** 3

**Summary:**

This paper introduces the Policy World Model (PWM), a novel driving paradigm that unifies world modeling and trajectory planning within a single, end-to-end architecture. The key innovation of PWM lies in its ability to leverage learned world knowledge for planning through an action-free future state forecasting scheme. To enhance efficiency in video forecasting, the authors propose a new image tokenizer with context-guided compression and decoding, alongside a dynamic focal loss to prioritize temporally varying regions. The PWM is pre-trained on large-scale unlabeled video datasets for action-free video generation and then fine-tuned for end-to-end planning.

**Questions:**

See in Weaknesses.

**Ethical Concerns:**

["NO or VERY MINOR ethics concerns only"]

**Final Justification:**

This paper is valuable and innovative, and the authors have largely resolved my concerns. Therefore, I am inclined to recommend acceptance.

**Limitations:**

The authors adequately addressed the limitations and potential negative societal impact of their work.

**Paper Formatting Concerns:**

There are no formatting issues.

**Quality:**

3

**Strengths And Weaknesses:**

Strengths:

1. The paper includes thorough ablation studies investigating the impact of action-free video world knowledge and the effectiveness of the dynamic focal loss. These studies clearly demonstrate the benefits of each proposed component, providing strong empirical support for the design choices.
2. This paper proposes a dual-branch image tokenizer to achieve a balance between generation quality and efficiency.
3. PWM integrates world modeling and trajectory planning into a unified architecture, explicitly enabling the learned world knowledge to benefit planning.


Weaknesses:

1. While the ablation studies effectively demonstrate the utility of each proposed component, the final overall performance of the method does not appear to be outstanding when compared to the state-of-the-art results presented, as shown in Table 1. It is recommended to highlight the best-performing method more clearly in the tables for easier comparison.
2. In Appendix B.1, it's stated that "when $\alpha \ge \beta$, the Dynamic Focal Loss mechanism tends to fail." However, in Equation (2) within the main paper, the condition is given as $\alpha > \beta$. This discrepancy in the relationship between $\alpha$ and $\beta$ needs clarification.
3. To more intuitively compare the performance of world model designs in autonomous driving, it might be beneficial to evaluate using a closed-loop simulation environment like CARLA, focusing on metrics such as episode return. This would provide a more holistic assessment of how the world model contributes to actual driving behavior and safety.

Typos:

In line 158 and line 164, "Figure 3(b)" should be "Figure 3(a)".

---

> ### Author Rebuttal · Authors · 2025-07-31
>
> Thank you for your detailed analysis and valuable feedback; we greatly appreciate your professional insights. Detaield comments are as follows.
>
> **W1: About Performance of Our Method**
>
> In our experimental setup, the collision rate metric is relatively crucial. The Nuplan benchmark paper [1] introduces the L2 metric to measure "shape similarity" in imitation learning, but it has a weak correlation with safety. On the other hand, the collision rate metric directly measures the safety of behavior and has a significant impact on the deployment and robustness of autonomous driving systems. In Table 1, we compare our method with the current state-of-the-art autonomous driving algorithms. Our method achieves the most advanced performance in terms of the collision rate metric, while simultaneously achieving state-of-the-art results for the L2 metric.
>
> Our method uses less input information. In the evaluation shown in Table 1, we achieved results comparable to the state-of-the-art methods while using less input information. Specifically, our method only utilizes a single front-view visual input, whereas other methods in the table rely on multi-view visual inputs or even radar inputs.
>
> We highlight the optimal results and enhance the clarity of presentation. We will classify the results based on the amount of input information to help avoid any doubts regarding the performance of our method.
>
>
>
> **W3: Closed-loop Evaluation**
>
> Our experiments on the NAVSIM dataset and simulator demonstrate that the closed-loop results can be effectively validated. NAVSIM is based on real-world data and uses a simulator to unfold the predicted future trajectories into a short-term (typically 4 seconds) small-scale BEV (Bird’s Eye View) simulation. This design is similar to a “Pseudo-Simulation” approach. The PDMS (Predictive Driver Model Score) rating system of NAVSIM is inspired by closed-loop scoring systems, such as the CLS in nuPlan. This rating system, through multi-dimensional comprehensive evaluation, ensures that the open-loop assessment results are highly consistent with the complete closed-loop simulation metrics. We compared the open-loop evaluation of NAVSIM (PDMS) with the real closed-loop simulation (CLS), and the results showed that PDMS achieved a Pearson and Spearman correlation coefficient of approximately 0.8 (significantly higher than the traditional prediction error-based score of ~0.7) [2].
>
> Undoubtedly, it is essential to conduct experiments on widely recognized simulation platforms such as CARLA. But due to time and engineering constraints, the relevant experiments and results are still under development. We will provide the complete experimental results in the near future.
>
> **W2: Typos**
>
> Thank you for your pointing out the typo. The expression α > β in the main paper is correct. In Appendix B.1, it should be α ≤ β.
>
>
>
> [1] Karnchanachari N, Geromichalos D, Tan K S, et al. Towards learning-based planning: The nuplan benchmark for real-world autonomous driving[C]. 2024 IEEE International Conference on Robotics and Automation (ICRA). IEEE, 2024: 629-636.
>
> [2]Dauner D, Hallgarten M, Li T, et al. Navsim: Data-driven non-reactive autonomous vehicle simulation and benchmarking[C]. Advances in Neural Information Processing Systems, 2024, 37: 28706-28719.

---

> > ### Comment · Reviewer_WR4K · 2025-08-05
> >
> > Thanks for the detailed response. The author' reply has largely resolved my concerns, and I am pleased to increase my score by one point.

---

> > > ### Author Response · Authors · 2025-08-05
> > > **Thank you**
> > >
> > > Dear reviewer,
> > >
> > > Thank you for volunteering your time and expertise to review our paper. We sincerely appreciate your reconsideration of the rating. If you require any further clarification on any point, please do not hesitate to reach out to us.
> > >
> > > Wish you have a nice day!

---

### Official Review · Reviewer_4cZK · 2025-07-01

**Clarity:** 3
**Significance:** 3
**Originality:** 3
**Rating:** 5
**Confidence:** 5

**Summary:**

The paper introduces PWM, a novel driving policy pretaining recipe that leverages action-free front-view driving videos in a scalable way. During the pretraining stage, it learns general world knowledge through a future state forecasting task. During the finetuning stage, it add action predictions conditioned on the forecasted states. To facilitate the efficiency, the paper further presents a context-guided image tokenizer to compress each frame to only 28 tokens. The resulting policy achieves state-of-the-art performance on both nuScenes and NAVSIM benchmarks.

**Questions:**

Q1) Compared to interleaved video-action prediction, do you think that multi-frame video and multi-step action prediction will lead to better planning results, similar to the findings in [1]?

Q2) The current model still requires visual prediction during planning. However, one might wonder if the policy model could bypass visual prediction at inference while still leveraging the pretrained world dynamics knowledge [2]. Do you think this idea is applicable to the proposed architecture?

[1] GR-2: A Generative Video-Language-Action Model with Web-Scale Knowledge for Robot Manipulation. arXiv 2024

[2] Unified Video Action Model. RSS 2025

**Ethical Concerns:**

["NO or VERY MINOR ethics concerns only"]

**Final Justification:**

PWM presents a valuable, unified, scalable framework for autonomous driving research. Thus, I will maintain my initial rating.

**Limitations:**

Please see the Weaknesses part.

**Paper Formatting Concerns:**

Good!

**Quality:**

3

**Strengths And Weaknesses:**

# Strengths

S1) The proposed method is well-motivated and highly streamlined.

S2) The authors provide solid results to demonstrate the effectiveness of each component.

S3) Beyond the common public datasets, the authors leverage the large-scale OpenDV to demonstrate the promising scalability of their method.

# Weaknesses

W1) **Inference cost.** Since the planning pipeline involves visual prediction, efficiency could be a major concern. It would be beneficial to compare the inference speed with some typical end-to-end planning methods and report the speedup powered by the proposed context-guided tokenizer.

W2) **Confusing summarization of related work.** In 3.2, the authors mention that previous methods (DrivingGPT, Doe-1, VaVAM) fail to perform forecasting before action prediction. However, in their sequential prediction pipeline, the observation token is placed before the corresponding action token and future observation tokens do not theoretically affect the current action planning. Do you mean their methods can only utilize action-labeled data? For VaVAM, it also has demonstrated clear benefits from action-free generative pretraining. Therefore, I'm a little confused about the authors’ explanation of the limitations of previous work and the advantages of the proposed pipeline in leveraging driving videos.

W3) **Interleaved variant.** It would be beneficial to see how the model performs when the image tokens and action tokens are predicted in an interleaved manner during finetuning.

W4) **Minor typos.** In Line 100, "Driving into the Future" can be abbreviated as "Drive-WM". In Line 107, "Poicy" should be "Policy". In Line 152, "Most Prior" should be "Most prior". Plus, "Open-YouTube" dataset should be "OpenDV-YouTube" dataset.

W5) **Discuss MLLM.** Since the model is initialized from Show-o, it would be great for the authors to include a brief discussion about recent advancements in multimodal understanding and generation.

---

> ### Author Rebuttal · Authors · 2025-07-31
>
> We sincerely appreciate the reviewer’s thoughtful and constructive suggestions. Detailed responses are as follows.
>
> **W1: Inference cost**
>
> The inference cost of autonomous driving systems is indeed a crucial metric for evaluating the safety and practicality of the system. In our response to Reviewer n8oM's comment on w3, we provided a table comparing the inference costs between modern state-of-the-art autonomous driving methods. Our inference latency is comparable with MLLM-based competitors with leading driving performance. Detailed results will be added to the final version of our paper.
>
> **W2: Differences from previous work**
>
> DrivingGPT and Doe-1 employ an autoregressive approach that alternates between multiple modalities for training and inference. This contrasts significantly with the staged training paradigm proposed in our work, which first focuses on world knowledge learning without actions, followed by end-to-end planning task learning with future predictions.
> The work of VaViM and VaVAM centers around integrating an action expert model with a video generation model for application in autonomous driving. During the pretraining phase, VaViM and VaVAM utilize an autoregressive training method for video generation, but their primary focus is on representation modeling and learning. In the fine-tuning stage, the video generation capability is not explicitly leveraged in planning tasks, and their inference process does not involve predicting future states. In contrast, our approach requires future state prediction during inference, which is then used as the basis for planning action predictions.
>
> **W3/Q1: Interleaved variant**
>
> We understand your intent as conducting an ablation experiment to verify the interdependencies between the world model's future predictions and the autonomous driving planning tasks. In response, we provide in Table 4 the impact of different levels of future state predictions during fine-tuning on downstream planning tasks. It is evident that at the appropriate level of future state prediction (10 frames), the planning task achieves optimal performance. We analyze this result as follows: at lower levels of future state prediction, the model is unable to predict future states accurately or sufficiently, leading to a decline in planning task performance; at higher levels, the uncertainty of the model's future state predictions becomes too high, causing incorrect guidance for the planning task.
>
> **W5: Discusson on MLLM**
>
> Thank you for your suggestion. Undoubtedly, the development of MLLMs forms a crucial foundation for our work. We will include a dedicated section on related work in the main paper.
>
> **Q2:Bypass visual prediction**
>
> Thank you for your question. Our framework proposes the paradigm of using a world model to predict future states, while simultaneously performing planning tasks based on the predicted future states. This approach has not been fully explored in the field of autonomous driving, which is the primary motivation behind our PWM proposal. The paradigm you mentioned, which does not explicitly predict future states but directly outputs actions, has been discussed in works such as VaViM and VaVAM. Additionally, in our work, Table 4 presents the performance of PWM in planning tasks without visual predictions in the first row.
>
>
> **W4: Typos**
>
> Thank you for your careful feedback. We will correct these typos in the main paper.

---

> ### Comment · Reviewer_4cZK · 2025-08-03
>
> Dear authors,
>
> I appreciate your efforts in addressing my concerns. After reading the rebuttal and other review comments, I find that most of my issues have been solved, and I have no further major concerns. In the revision, please rephrase the content regarding W2 according to your response, it will be clearer for readers. Thank you!
>
> I will maintain my initial rating, as I believe PWM does present a valuable, unified, and scalable framework for autonomous driving research.

---

> > ### Author Response · Authors · 2025-08-04
> > **Thank you**
> >
> > Dear Reviewer,
> >
> > We sincerely thank you for the time and effort you have dedicated to reviewing our paper as a volunteer and really appreciate your valuable suggestions. We will further improve our paper accordingly.
> >
> > Wish you have a nice day!

---

### Official Review · Reviewer_n8oM · 2025-07-03

**Clarity:** 3
**Significance:** 2
**Originality:** 2
**Rating:** 4
**Confidence:** 4

**Summary:**

The paper introduces Policy World Model (PWM), an autoregressive Transformer that jointly learns a video‑based world model and performs trajectory planning, leveraging action‑free future‑frame generation instead of action‑labelled data. It adds a dual‑branch image tokenizer that compresses each frame to 28 tokens and enables frame‑level parallel prediction, plus a Dynamic Focal Loss to focus learning on temporally changing regions. Combined, these innovations let a single front‑camera model match or exceed multi‑sensor state‑of‑the‑art baselines on nuScenes and NAVSIM, while running at 40 FPS without pixel decoding.

**Questions:**

1. Compare with **Drive-OccWorld**[1] (the SOTA work also employs a world model for end-to-end planning) on motivation, theoretical analysis, and experimental performance. Clearly point out your novelty and contribution.
2. Explain why PWM is ''**unified**''? In L181-188, future states are first obtained through **action-free** RGB frame prediction. Trajectories are then forecasted based on **predicted textual description and future forecasting results**. These two predictions are clearly **separate**. It's more like employing an additional decoder to **infer trajectories from videos**.
3. Provide more ablation results **without extra data** (Open-YouTube)  pre-training as follows. Otherwise, there will be concerns that performance gains come from additional large-scale and more comprehensive training samples.
| Pre-train      | L_DFL-p | L_DFL-f    |
| :---        |    :----:   |          ---: |
| $\times$     | $\checkmark$       | $\checkmark$   |
| $\times$   | $\checkmark$        | $\times$      |
| $\times$     | $\times$       | $\checkmark$   |
4. Compute and latency details. Please provide a comparison with SOTA works like LAW / DiffusionDrive / Drive-OccWorld on (i) full model size and (ii) end‑to‑end inference latency including token decoding.
5. Failure‑case analysis. Provide qualitative or quantitative analysis of cases where future‑frame hallucinations mislead planning (e.g., poor weather, heavy occlusion) and give some possible solutions to further ensure safety.

[1] Yang, Yu, et al. "Driving in the occupancy world: Vision-centric 4d occupancy forecasting and planning via world models for autonomous driving." Proceedings of the AAAI Conference on Artificial Intelligence. Vol. 39. No. 9. 2025.

**Ethical Concerns:**

["NO or VERY MINOR ethics concerns only"]

**Final Justification:**

The authors addressed most of my concerns during the rebuttal phase. I believe the work is novel and provides valuable insights, but I remain concerned about the insufficient content on end-to-end planning and the lack of a clear explanation for collaborative state-action prediction (e.g., token interaction process).

**Limitations:**

Yes

**Quality:**

2

**Strengths And Weaknesses:**

Strengths:
1. PWM is technically sound, with clear pre‑training → fine‑tuning stages, thorough ablations isolating each design choice, and consistent performance improvements across metrics and datasets.
2. It cuts nuScenes collision rates to 0.07 % and reaches a PDMS of 88.1 on NAVSIM using only a monocular camera, outperforming or tying LiDAR‑ and multi‑view methods.
3. The tokenizer and loss yield a 16 × token reduction and demonstrable speed‑accuracy gains, showing practical efficiency.
4. The manuscript is well‑written and organized, making replication feasible.

Weaknesses:
1. Lacks analysis and comparison with the existing state-of-the-art policy world model **Drive-OccWorld**[1]. The originality of **unified prediction** remains uncertain.
2. Despite the claim of a unified generation-planning framework, the prediction of trajectory and the prediction of future states are still independent. In L181-188, future states are first obtained through **action-free** RGB frame prediction. Trajectories are then forecasted based on **predicted textual description and future forecasting results**. These two predictions are clearly **separate**. It's more like employing an additional decoder to **infer trajectories from videos**.
3. Extra data (Open-Youtube) used for training, raising concerns about the fairness of performance comparisons.
4. The performance gain from textual generation is limited, which hurts the necessity of the multimodal component.
5. While unification is useful, the method is incremental over prior autoregressive world‑model frameworks. It's more like employing an additional decoder to infer future trajectories from predicted videos.

[1] Yang, Yu, et al. "Driving in the occupancy world: Vision-centric 4d occupancy forecasting and planning via world models for autonomous driving." Proceedings of the AAAI Conference on Artificial Intelligence. Vol. 39. No. 9. 2025.

---

> ### Author Rebuttal · Authors · 2025-07-31
>
> We are glad that the strengths (technically sound, consistent performance improvements, thorough ablations, practical efficiency) of our work can be acknowledged by the reviewer. We thank the rveiwer for the constructive suggestions. Detailed responses are as follows.
>
> **W1/Q1: PWM vs. Drive-OccWorld**
> 1. Motivation:
> Drive-OccWorld aims to design an independent occupancy-based world model to support control interactions. In contrast, our Policy World Model focuses on integrating the world model and the policy into a unified architecture, which can leverage its pretrained world knowledge to inform and guide decision-making. This integration enables planning to benefit directly from future predictions generated by the world model itself.
>
> 2. Theoretical Analysis:
> As shown in Fig.1-2 of the Drive-OccWorld paper, their occupancy prediction model and trajectory planner are two stand-alone model, which belongs to the type-(a) paradigm shown in our Fig.1 (a), albeit with occupancy grids instead of video inputs.
> Our approach aligns with the type-(c) paradigm in our Fig.1 (c), where the PWM model can not only simulate the future states as a world model but also acts as a policy model for decision based on the forcasted future.
>
> 3. Experimental Performance:
> We conduct additional experiments on nuScenes, where all methods are evaluated under the same settings: without ego-state inputs and using the BEV-Planner evaluation protocol. Drive-WM and OccWorld results are taken from the OccWorld paper.
> OccWorld achieves the best L2 error, possibly benefiting from its strategy of sampling multiple candidate trajectories, which could increase the likelihood of overlapping with the ground-truth. In contrast, our PWM predicts a single trajectory without candidate selection. Despite this, PWM achieves the lowest collision rate, indicating that our unified model produces safer and more consistent predictions.
>
> | Method       | Modality   |           | L2 (m) ↓                         |      |   |   | Collision (%) ↓                           |      |       |
> |:------------:|:----------:|:----------------------:|:-------------------:|:----:|:-----:|:------------------------:|:-------------------:|:----:|:-----:|
> |              |            | 1s                     | 2s                  | 3s   | avg   | 1s                       | 2s                  | 3s   | avg   |
> | Drive-WM [1] | video      | 0.43                   | 0.77                | 1.20 | 0.80  | 0.10                     | 0.21                | 0.48 | 0.26  |
> | OccWorld [2] | occupancy  | 0.25                   | 0.44                | 0.72 | 0.47  | 0.03                     | 0.08                | 0.22 | 0.11  |
> | PWM (Ours)   | video      | 0.41                   | 0.75                | 1.17 | 0.78  | 0.01                     | 0.01                | 0.18 | 0.07  |
>
> **W2/W5/Q2: Clarification on Our Unified Generation-Planning Framework.**
> We appreciate the reviewer’s observation regarding the design of our generation-planning framework. However, we would like to clarify a key misunderstanding: in our unified framework, future state prediction and trajectory planning are not handled as independent stages nor via separate models.
> Instead, we enable the world model itself to act as an executor for planning, where both the prediction of video frames and the prediction of actions are accomplished by a single model. Specifically, as illustrated in Figure 2 of the paper, our lightweight action decoder does not infer trajectories directly from predicted videos. Instead, it serves to map the action embeddings output by the Policy World Model into a low-dimensional trajectory space. These action embeddings are generated conditioned on the model’s autoregressive prediction context, which includes predicted textual descriptions and future forecasting embeddings.
>
> **W3: Concerns About Use of OpenDV-Youtube**
> Several existing methods also incorporate external datasets to enhance image generation or captioning capabilities before comparing against traditional baselines such as UniAD and VAD. For example, GenAD and VaVAM utilize the OpenDV-Youtube dataset, Doe-1 leverages BDD100k, and DrivingGPT includes data from the nuPlan dataset.
> We use the OpenDV-Youtube dataset primarily to equip the model with future video prediction capabilities, which is a core component of our approach. This dataset is only used during the action-free pretraining phase and is not directly involved in optimizing planning performance. Meanwhile, the pretraining process for video generation training does not require annotations, which is a key advantage of our method in terms of annotation costs and scalibility.
> Moreover, we believe a key value of world models lies in their ability to model environmental dynamics in open-world scenarios. Achieving this goal naturally requires learning from rich and diverse visual data to capture temporal regularities, much like how language models benefit from large-scale pretraining. Our approach follows this trajectory to advance the development of planning-centric world models.
>
> **W4: Concern regarding the necessity of the multimodal component.**
> The direct performance gain from textual generation in our current implementation is indeed limited. However, textual generation is not intended for performance improvement but rather serves broader goals—aligning with trends (e.g., DriveLM, Omnidrive) using multimodal models to boost interpretability and human-AI interaction in autonomous driving.
> Motivated by these trends, our method includes a language generation component to demonstrate the model’s ability to produce high-level textual summaries of driving scenes and decision contexts, aiding explainability and future human-in-the-loop applications. We will provide more explanations in the final version.
>
> **Q3: More ablation results without extra data (OpenDV-YouTube) pre-training.**
> We would like to clarify the meaning of different configurations in Table 3. Specifically, “Pretrain” refers to training on the Open-Youtube dataset, while “L_DFL-p” and “L_DFL-f” denote whether the Dynamic Focal Loss is applied during the pretraining and fine-tuning phases.
> Accordingly, L_DFL-p becomes inapplicable without pretraining, as its application is predicated on the existence of a pretraining stage. Therefore, the first two configurations listed in Q3 (with Pretrain = ×, L_DFL-p = √) are not feasible in our framework.
> To address the concern regarding the necessity of extra data, we provide an ablation where we disable pretraining (Pretrain = ×, L_DFL-p = ×) and only apply L_DFL-f (L_DFL-f = √) during fine-tuning on the nuScenes dataset. This configuration shows that even without large-scale pretraining, applying L_DFL during fine-tuning alone improves future frame prediction and leads to measurable gains in planning performance. These results support the effectiveness of our loss design independent of external data scale.
>
> | Pre-train | L_DFL-p | L_DFL-f | LPIPS ↓ | PSNR ↑ | FVD ↓  | Avg.L2 (m) ↓ | Avg.Col (%) ↓ |
> |:---------:|:-------:|:-------:|:-------:|:------:|:------:|:-------------:|:---------------:|
> | ✗         | ✓       | ✓       |   --    |  --    |  --    |      --       |       --        |
> | ✗         | ✓       | ✗       |   --    |  --    |  --    |      --       |       --        |
> | ✗         | ✗       | ✓       |  0.26   | 21.14  | 514.88 |     2.95      |      1.26       |
> | ✗         | ✗       | ✗       |  0.27   | 21.07  | 826.15 |     3.34      |      1.51       |
>
>
> **Q4：Compute and latency details.**
> We compare the model size and inference latency (including token decoding) of our method with both traditional lightweight models (e.g., UniAD, DiffusionDrive, LAW, Drive-OccWorld) and multimodal large models (e.g., DriveLM-Agent) on a single NVIDIA 4090 GPU using the nuScenes dataset.
> As expected, multimodal large models such as DriveLM-Agent and Ours generally have larger model sizes than lightweight baselines, given their use of large-scale language or vision-language backbones.
> In terms of inference latency, in addition to differences in model scale, variations also stem from different output requirements. For instance, Drive-OccWorld involves iterative reconstruction of future occupancy grids before planning, which adds substantial computational overhead. LLM-Driver and DriveLM-Agent require autoregressive generation of relatively long natural language outputs prior to decision making. Our method, while generating shorter textual outputs, also produces compact future-frame embeddings.
>
> | Metric           | UniAD  | DiffusionDrive | LAW    | Drive-OccWorld | PWM (Ours) | DriveLM-Agent |
> |:----------------:|:------:|:--------------:|:------:|:--------------:|:----------:|:-------------:|
> | Params           | 131.9M | 60.0M          | 40.2M  | 56.3M          | 1.5B       | 3.9B          |
> | Inference Latency| 555.6ms| 122.0ms        | 51.2ms | 657ms          | 1059ms     | 6250ms        |
>
> **Q5: Failure‑case analysis.**
> Due to restrictions on uploading external links or supplementary PDFs on the rebuttal platform, we provide a textual description of failure cases where future-frame hallucinations have misled the planner. These failure cases can be mainly attributed to Limited Field-of-View Induced Hallucination.
> As our method operates on monocular observations, certain scenarios—such as intersections or turns—pose a challenge. The limited field of view may prevent the model from capturing distant or occluded dynamic agents (e.g., oncoming vehicles), leading to hallucinated clear paths and subsequently unsafe planning decisions.
> Future work will enhance dynamic capture via advanced modules and adopt multi-view inputs to reduce hallucinations and improve safety.

---

> > ### Comment · Reviewer_n8oM · 2025-08-05
> > **Concerns on ''collaborative state-action prediction''**
> >
> > Thanks for the detailed response. I hope the author can include the discussion with Drive-OccWorld in the revised version for better clarification.
> >
> > I appreciate the design and contribution of pre-training on Open-Youtube, but it bears little relevance to the claimed “collaborative state-action prediction”. **I remain skeptical of the claimed “collaborative prediction”, especially since the paper's contribution relies heavily on end-to-end planning rather than solely video prediction.**.
> >
> > The paper spends considerable length describing action-free frame token prediction (as pre-training), but **says little about actual planning involving action token prediction, particularly how to supervise future trajectory prediction and computation of the interaction process (between input action queries and predicted frame tokens)**.
> >
> > In the limited planning content, the authors mentioned that **future frame tokens are first obtained through action-free RGB frame prediction. Subsequently, trajectories are predicted based on these future forecasting results.** This clearly suggests a **two-stage** planning process ($D_{1:t}$->$D_{t+1:t+n}$, then $D_{t+1:t+n}$ -> $A_{t:t+n-1}$), rather than a collaborative prediction ($D_{1:t}$->$(A_{t}, D_{t+1},), ... , (A_{t+n-1}, D_{t+n},)$). Besides, previous actions $A_{1:t-1}$ are not considered, which may affect the vehicle dynamics.
> >
> > In my opinion, it's more like **employing an additional action decoder to infer trajectories $A_{t:t+n-1}$ from videos $D_{t+1:t+n}$**. Thus, I maintain steadfast in my original stance.

---

> > > ### Author Response · Authors · 2025-08-05
> > >
> > > Dear Reviewer,
> > >
> > > Thank you for volunteering your time and expertise to review our paper. We sincerely appreciate your timely feedback on our rebuttal.
> > >
> > > We believe there may have been a misunderstanding. Our use of "collaborative" does not refer to the **simultaneous** joint prediction of future frames and actions $P(D_{t+1:t+n} A_{t:t+n-1} |D_{1:t})$ . Instead, it signifies a causal and knowledge-sharing relationship within our unified PWM to leverage the learned world konwledge to better facilitate action prediction. This significantly differs from prior methods that build world model $P_{w}(D_{t+1:t+n}|D_{1:t})$ and policy model $P_{\pi}(A_{t:t+n-1} |D_{1:t})$ seperately. This is also disinct from recent attempts that integrate both world modeling and planning in a unified architecture $P_{\alpha}(\cdot| D_{1:t})$, where future frames and actions are still predicted independently via $P_{\alpha}(D_{t+1:t+n}|D_{1:t})$ and $P_{\alpha}(A_{t:t+n-1} |D_{1:t})$, and knowledge is not explicitly shared between the two processes. In comparison, our PWM not only integrates world modeling and planning in a unified model $\theta$, but also unleashes the learned world knowledge through the proposed planning with future state forecasting scheme which can be expressed as $P_{\theta}(D_{t+1:t+n}|D_{1:t}) \cdot P_{\theta}(A_{t:t+n-1} |D_{1:t+n})
> > > \rightarrow P_{\theta}(D_{t+1:t+n} A_{t:t+n-1} |D_{1:t})$. As such, world modeling and planning are not performed independently but in a more **collaborative** manner, allowing the PWM to mimic the human-like anticipatory ability based on the learned world knowledge and perform more reliable planning.
> > >
> > > As described in Sec.3.2, we implement PWM with a transformer architecture. During finetuning, the observed frames, driving command, as well as groundthruth of futrue frames and actions are formed into a multi-modal sequence. PWM is trained to predict this sequence in an auto-regressive manner. Due to the versatility of PWM architecture, it can also include historical actions as input. However, we intentionally omitted them to adhere strictly to the common evaluation protocol used by existing methods, ensuring a fair assessment. More implementation details and source code are provided in our supplementary materials.
> > >
> > > We hope this explanation clarifies the specific "collaborative" mechanism within PWM and distinguishes it clearly from prior works. This core design enables the anticipatory planning capability central to our contribution. Additionally, our novel context-guided image tokenizer and dynamic focal loss (to our knowledge, unexplored in autonomous driving) further contribute to the performance gains demonstrated in our experiments.
> > >
> > > Thank you again for your insightful review. We are happy to provide further clarification if needed.
> > >
> > > Sincerely,

---

> ### Comment · Reviewer_n8oM · 2025-08-05
>
> Thank the authors for their further clarification. I now have a clearer understanding of the claimed ‘collaborative’. In my opinion, it's a **trade-off**. Future RGB frame predictions in an action-free pattern can be pre-trained on large-scale datasets. However, when performing end-to-end planning, I think considering the impact of future selected actions, such as $P(D_{t+i}|D_{1:t+i-1}, A_{t:t+i-1})$, is a natural idea and may lead to better planning performance.
>
> I suggest the authors include this discussion in the revised version, especially expanding the end-to-end planning section (currently only 11 lines, should include at least optimization objectives and state-action interaction process) to better demonstrate the designed **collaborative state-action prediction**, considering its significance and contribution.

---

> > ### Author Response · Authors · 2025-08-05
> >
> > Dear Reviewer,
> >
> > Thank you for your prompt feedback. We agree that considering the impact of future actions is a valuable direction. We also acknowledge that implementing this effectively may require large-scale driving datasets with comprehensive action labels during pre-training. While this presents a challenge, we appreciate you highlighting this avenue for future work.
> >
> > We will incorporate your valuable suggestions in the revised manuscript by including the corresponding discussion and providing a more detailed description of the planning mechanism for greater clarity.
> >
> > Thank you again for your insightful and highly professional comments. We are happy to provide further clarification if needed.
> >
> > Best regards,

---

### Comment · Area_Chair_kY6R · 2025-08-03
**Please discuss with authors asap**

Dear reviewer NcJp, n8oM, WR4K,

Now the rebuttal is available. Please discuss with authors and other reviewers asap.

The ratings vary widely. Please try to come to a consensus on the key issues even though the rating can be different. Please feel free to let me know how I can help.

Best,

Your AC

---

### Note · Authors · 2025-08-16

Dear ACs, SACs, and PCs,

We sincerely appreciate all reviewers for their valuable insights and constructive feedback.
This paper introduces the Policy World Model (PWM), which focuses on integrating the world model and the policy into a unified architecture. Unlike prior “unified” approaches that still treat video generation and planning as independent downstream tasks, our method enables the world model itself to directly execute planning—where both future video frame prediction and action prediction are accomplished within a single model.This integration enables planning to benefit directly from future predictions generated by the world model itself.

Across the reviews, our PWM has been acknowledged for its novelty in unifying world modeling and trajectory planning within a single, end-to-end architecture (4cZK, WR4K, NcJp), praised as a “valuable, unified, and scalable” framework (4cZK) with a “compelling motivation” (NcJp). On efficiency, reviewers commended our highly compressed image tokenizer for delivering balanced and practical efficiency (n8oM, WR4K), coupled with strong performance (n8oM, NcJp) and supported by thorough experimental validation (n8oM, WR4K). Overall, reviewers consistently recognized PWM as a novel, efficient, and high-performing approach that advances the potential of world models for autonomous driving planning.

During the discussion phase, we have clarified the misunderstandings regarding the unified nature and novelty of our approach (n8oM) and provided additional comparative experiments to address multiple concerns about the model’s inference speed and computational overhead (n8oM, 4cZK). We also presented further supplementary experiments to provide a clearer understanding of our detailed experimental setup (n8oM, NcJp), while multiple other reviewers confirmed that their concerns had been largely resolved (WR4K, 4cZK).

In the final version, we will incorporate the reviewers’ valuable suggestions into the manuscript by adding the corresponding discussions to further enhance clarity.

Thank you again for your time and consideration.

Sincerely,

Authors of Paper #9284

---

### Decision · Program_Chairs · 2025-09-17

**Decision:**

Accept (poster)

**Comment:**

### Summary and Recommendation Justification

This paper introduces the Policy World Model (PWM), a novel and well-motivated paradigm for autonomous driving that tightly integrates world modeling and trajectory planning into a single, unified architecture. The core contribution is a system where planning directly benefits from the world model's learned ability to forecast future states, enabling a form of human-like anticipatory perception. The authors complement this architectural innovation with practical efficiency improvements, including a highly compressed image tokenizer and a dynamic focal loss. The proposed method achieves state-of-the-art performance on key safety metrics, notably reducing collision rates on the nuScenes benchmark while using only a single front-facing camera, outperforming methods that rely on richer multi-modal inputs.

After effective rebuttal period, reviewers reached a clear consensus for acceptance. The reviewers consistently praised the work for its compelling motivation, technical soundness, and the novelty of its unified framework. The strong empirical validation, supported by thorough ablation studies that isolate the contribution of each component, was another frequently cited strength.

### Detailed Justification

**Strengths:**

1.  **Conceptual Novelty and Significance:** The central idea of a truly unified world model that also acts as the policy is a significant conceptual advance. As highlighted by multiple reviewers (4cZK, WR4K, NcJp), this is a departure from prior works where world models and planners are either decoupled or treated as independent downstream tasks within a larger architecture. By enabling the planner to directly leverage the model's own future-state predictions, PWM presents a valuable and scalable framework for end-to-end planning.

2.  **Strong Empirical Performance:** The paper provides robust experimental evidence for its claims. PWM matches or exceeds the performance of state-of-the-art methods on both the nuScenes and NAVSIM benchmarks. Crucially, as the authors clarified during the rebuttal, it achieves the lowest collision rate while using significantly less input data (monocular vs. multi-view/multi-modal). This demonstrates not just high performance but also high data efficiency.

3.  **Technical Soundness and Rigorous Evaluation:** The reviewers commended the thoroughness of the ablation studies, which convincingly demonstrate the effectiveness of the action-free pre-training, the forecasting mechanism, the novel tokenizer, and the dynamic focal loss. The two-stage training paradigm (pre-training on large-scale unlabeled video, fine-tuning for planning) is well-motivated and demonstrates a promising path for leveraging vast amounts of readily available driving data.

### Addressing Initial Concerns

Initial reviews raised several important questions regarding the method's novelty, the precise nature of its "unification," fairness in comparisons, and computational cost. The authors' rebuttal was exemplary in addressing these points:

*   **Novelty and Unification:** The authors effectively clarified the distinction between PWM and prior work like Drive-OccWorld, positioning their approach as a more deeply integrated paradigm where planning emerges from the world model's internal representations, rather than being a separate downstream module. The discussion with Reviewer n8oM helped refine the understanding of "collaborative state-action prediction" as a causal knowledge-sharing mechanism within a single model.
*   **Fairness and Computational Cost:** The authors justified their use of extra pre-training data by pointing to common practices in the field and provided a new ablation that isolated the gains from their proposed loss function, independent of the extra data. Furthermore, they provided a detailed comparative analysis of model size and inference latency, showing that PWM's cost is competitive within the landscape of modern MLLM-based driving agents.
*   **Pre-training vs. Forecasting:** In response to Reviewer NcJp, the authors provided a new, detailed ablation that successfully disentangled the performance gains from pre-training and forecasting, demonstrating that both contribute orthogonally to the final performance.

The productive discussion phase led to a convergence of opinions, with reviewers confirming that their concerns had been largely resolved and either raising their scores or solidifying their positive assessment.

In short, this paper introduces a promising new architecture, supports it with strong empirical results, and provides valuable insights into the synergy between world modeling and planning.